# Multipole Semantic Attention: A Fast Approximation of Softmax Attention for Pretraining

**Rupert Mitchell** [1]   **Kristian Kersting** [1 2 3 4]

## Abstract

Pretraining transformers on long sequences, such as entire code repositories or collections of related documents, is bottlenecked by quadratic attention costs. We present Multipole Semantic Attention (MuSe), which accelerates 64k-context pretraining by 36% while matching baseline loss, requiring no architectural changes. MuSe is a training-time approximation that clusters queries and keys separately in representation space. This yields query-specific summaries that substantially outperform spatial blocking at matched sparsity, while also enabling drop-in compatibility with existing pretrained models—we validate on Llama 3.1-8B and 3.2-1B without retraining. We pretrain language models up to 1B parameters at 64k context on code and scientific documents, confirming that MuSe preserves quality and long-context utilization during training.

## 1. Introduction

The quadratic computational complexity of softmax attention remains the primary bottleneck limiting context length in transformers. While this $\mathcal{O}(N^2 D)$ scaling enables the rich token interactions that underpin transformer capabilities, it renders long-context pretraining prohibitively expensive. Modern pretraining increasingly benefits from extended context—whether processing entire code repositories or collections of related scientific papers—yet computational constraints force most models to train on artificially truncated sequences. Modern transformers partially mitigate these issues through the use of flash attention (Dao et al., 2022), which reduces memory complexity to linear

but retains quadratic computational complexity. Hybrid architectures using sliding window attention for local interactions must still interleave quadratic-complexity global attention layers to maintain full-context understanding. Efficient approximations to global attention, which maintain training quality at these scales, become essential for practical long-context pretraining.

In this context, we present Multipole Semantic Attention (MuSe), combining semantic clustering with ideas from computational physics to approximate softmax attention at training time. We demonstrate that (1) the approximation matches or exceeds baseline training quality, (2) throughput improvements of 36% are achievable with current hardware at context lengths of 64k, and (3) the method integrates into existing training pipelines without architectural changes. More specifically, MuSe introduces three key elements: First, we cluster queries and keys separately in their learned representation spaces, enabling a two-stage mechanism where coarse query clusters attend to fine key clusters, then fine queries refine through cluster summaries. Second, we augment centroid-based (monopole) approximations with retrieval of the most relevant clusters for exact attention. Third, query centroids provide exponential tilts that center the approximation around each query cluster's region of attention, enabling drop-in compatibility with existing pretrained models without retraining.

This distinction between the key and query latent spaces, whereby we cluster keys and queries independently, is motivated as follows. Firstly, softmax attention is invariant under translation of the keys (as this change is absorbed by the normalizing constant), but not of the queries. More importantly, the queries live in the dual vector space to the keys, that is, softmax attention is further invariant to an arbitrary change of basis of the keys, so long as the queries are transformed inversely. The consequence of this is that, unless one has some way to choose a preferred basis, it is unclear what it would mean to say that some particular key and query occupied the same point in latent space.

We validate MuSe empirically through microbenchmarks and end-to-end pretraining at scale. On isolated attention layers, we achieve $2\times$ speedup compared to CUDNN Flash Attention at 64k context with approximation errors below

---

[1]Department of Computer Science, TU Darmstadt, Darmstadt, Germany [2]Hessian Center for AI (hessian.AI), Darmstadt, Germany [3]German Research Center for Artificial Intelligence (DFKI), Darmstadt, Germany [4]Center for Cognitive Science, TU Darmstadt, Darmstadt, Germany. Correspondence to: Rupert Mitchell <rupert.mitchell@tu-darmstadt.de>.

*Proceedings of the 43$^{rd}$ International Conference on Machine Learning*, Seoul, South Korea. PMLR 306, 2026. Copyright 2026 by the author(s).

1%. We pretrain 1B parameter models on 64k context using both code repositories and scientific documents, demonstrating 36% wall-clock speedup while matching baseline quality. We further validate that the method generalizes to existing pretrained models (Llama 3.1-8B and 3.2-1B) and confirm that our models utilize the full 64k context during training.

To summarize, this paper makes the following contributions:

- A bidirectional semantic clustering approach that partitions queries and keys *separately* in their native representation space, enabling a hierarchical two-stage attention mechanism: coarse query clusters first attend to all fine key clusters to produce query-dependent summaries, then fine queries refine these summaries using their residual components

- A retrieval-augmented approximation that selects the most relevant key-value clusters for exact attention, with causal accumulation of summaries across spatial blocks

- Empirical validation at 1B parameter scale on code and scientific documents with 64k context, achieving 36% wallclock speedup while maintaining training quality

## 2. Related Work

We organize existing efficient attention methods into two broad categories: those which restrict attention to subsets of tokens exactly, and those which approximate attention globally. We fall into the latter category, and together with Hooper et al. (2025a) are the only works using semantic clustering therein.

**Restricted Attention**  Reformers (Kitaev et al., 2020) bucket queries and keys according to random directions and restrict attention to within buckets. Routing Transformers (Roy et al., 2021) learn a k-means clustering during training and similarly restrict attention to within buckets. Since late 2024 there has been significant work on K-means clustering in key-space for inference acceleration: Hooper et al. (2025b) use context-specific clustering to perform exact attention on the most relevant keys, while Tactic (Zhu et al., 2025) adapts the number of retrieved tokens to attention temperature.

**Approximate Attention**  Wang et al. (2020) use random projections to approximate the attention matrix, exploiting its low rank structure. Nyströmformer (Xiong et al., 2021) constructs a low rank representation using segment-wise means as landmarks. Fast Multipole Attention (Kang et al., 2023) and H-Transformer (Zhu & Soricut, 2021) use recursive spatial decomposition inspired by Barnes-Hut simulation (Barnes & Hut, 1986). Performer (Choromanski et al.,

*Figure 1.* MuSe method overview, depicting the far-field approximation. **Bars:** Queries (top) and keys/values (bottom) are partitioned by semantic cluster (A/B for queries, X/Y for keys) within each spatial block (0, 1, 2); segments vary in size because cluster membership is data-dependent. **Lower cube:** Per-block summaries indexed by (query cluster, key cluster, spatial block), constructed in Steps 1–2. **Upper cube:** Causally accumulated summaries (Step 3), accumulated along the spatial axis. **Steps 4–6:** A query in segment B2 (blue line) attends to the upper cube to select key cluster Y (Step 4, semantic selection), then to the lower cube to select spatial block 1 (Step 5, spatial selection), and finally attends exactly to keys in (Y, 1) (Step 6). Greyed-out segments do not participate: the first query block has no prior context, and the final key/value block is only accessed via local attention (computed separately). See Algorithm 1 for pseudocode.

2021) uses random features to approximate the exponential kernel. Mixture of Blocks Attention (MoBA; Lu et al., 2025) uses fixed spatial blocks without semantic clustering.

**Positioning of MuSe**  Our work uses semantic clustering of key-query representation space. Unlike sparse training methods such as MoBA that attend to fixed spatial blocks, we compute summaries of all clusters and augment with retrieval of high-attention clusters for exact computation. Unlike Hooper et al. (2025a) who apply monopole sum-

maries at inference time with key clustering only, we target pretraining where massively parallel queries make query clustering practical: computing query-specific summaries amortizes well while substantially improving approximation quality. Crucially, MuSe requires no architectural changes: the approximation is compatible with pretrained models (validated on Llama) and MuSe-trained models switch to exact attention with minimal adaptation.

## 3. Multipole Semantic Attention (MuSe)

Softmax attention is quadratic because every query attends to every key. The core idea of MuSe is to group both queries and keys by "semantic" similarity—similarity in the post-position-encoding representation space seen by attention—rather than solely by sequence position (key clusters depicted as green circles in Figure 2). Clustering queries separately from keys enables query-specific summaries that tightly approximate attention while remaining interchangeable with exact attention at test time. We combine this with spatial blocking for causality and selective retrieval of important (cluster, block) pairs for exact attention.

The core data structure is a summaries tensor of shape $C_q \times C_k \times (N/B)$, where $C_q$ and $C_k$ are the number of query and key clusters (typically $C_q = C_k = C$), and $N/B$ is the number of spatial blocks. Each entry summarizes the keys and values in a specific (query cluster, key cluster, spatial block) triple, enabling efficient approximate attention with selective exact retrieval.

**Spatial blocks** The context of length $N$ is divided into $N/B$ spatial blocks of size $B$ (e.g., 8 blocks of 8k tokens for 64k context). This structure enables causal masking at block granularity and provides the second level of retrieval.

**Clustering** Queries and keys are clustered *globally* across the full context[1] using a single pass of mini-batch K-means followed by a final assignment (see Appendix C), ignoring block boundaries. Each query $q$ decomposes as $q = \bar{q}_i + \tilde{q}$ where $\bar{q}_i$ is the centroid of its assigned cluster $i$ and $\tilde{q}$ is the residual. Clustering queries separately from keys—rather than clustering only keys as in prior work—is a central contribution.

Standard softmax attention computes, for each query $q$:

$$Z = \sum_k \exp(q \cdot k) \tag{1}$$
$$o = Z^{-1} \sum_k \exp(q \cdot k)\, v_k \tag{2}$$

As we show next, MuSe approximates the majority of these exponential weights by dropping the interaction between

---

[1]Clustering and retrieval selection are recomputed from the current queries and keys on every forward pass; MuSe maintains no state across sequences or training steps.

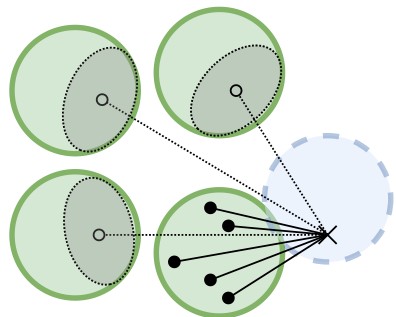

*Figure 2.* Geometric interpretation of MuSe. Green circles represent key/value clusters; the blue dashed circle represents a query cluster. Hollow dots mark the exponentially-tilted centroids—cluster summaries shifted toward the query cluster. Solid dots mark individual key/value pairs selected for exact retrieval. The chosen query (×) connects to tilted centroids via dotted lines (approximate attention) and to retrieved tokens via solid lines (exact attention).

query and key residuals. The result is accurate provided that any weights carrying large fractional error contribute negligibly to it.

**Why separate query clustering helps?** Consider the attention weight $\exp(q \cdot k)$ with $q = \bar{q} + \tilde{q}$ and $k = \bar{k} + \tilde{k}$. Expanding the dot product: $\exp(q \cdot k) =$

$$\exp\big(\bar{q}\cdot\bar{k}+\bar{q}\cdot\tilde{k}+\tilde{q}\cdot\bar{k}+\tilde{q}\cdot\tilde{k}\big) \approx \exp\big(\bar{q}\cdot\bar{k}+\bar{q}\cdot\tilde{k}+\tilde{q}\cdot\bar{k}\big) \tag{3}$$

If we cluster only keys (approximating $k \approx \bar{k}$), we lose the two terms involving $\tilde{k}$. With both query and key clustering, our two-stage mechanism retains three of four terms, dropping only $\tilde{q} \cdot \tilde{k}$—the product of two residuals. On our 1B model, this tighter approximation reduces monopole error by 3.7–5.6× (Table 3). Table 2 confirms both that query clustering provides a ∼9× effective cluster count advantage, and that finer clustering consistently reduces error by shrinking residual magnitudes.

### 3.1. Summary Computation and Causal Accumulation

We compress keys and values into compact cluster summaries—the monopole approximation—specialized for each query cluster via exponential tilting. These are then accumulated causally across spatial blocks.

**Per-block summaries** For each spatial block $b$, we compute summaries for all (query cluster $i$, key cluster $j$) pairs by attending from the query centroid $\bar{q}_i$ to all keys in block $b$ that belong to cluster $j$. Let $S_{ijb} = \sum_{k \in (b,j)} \exp(\bar{q}_i \cdot k)$ denote the unnormalized attention mass. The summaries

are:

$$\overline{k}_{ijb} = S_{ijb}^{-1} \sum_{k \in (b,j)} \exp(\overline{q}_i \cdot k)\, k \qquad (4)$$

$$\overline{v}_{ijb} = S_{ijb}^{-1} \sum_{k \in (b,j)} \exp(\overline{q}_i \cdot k)\, v \qquad (5)$$

$$\overline{\mu}_{ijb} = \log S_{ijb} \qquad (6)$$

These are "exponentially tilted" centroids—weighted by attention from the query centroid, not simple averages (shown as hollow dots in Figure 2, shifted toward the query cluster). This constitutes exact attention from coarse queries to fine keys. The cost is $\mathcal{O}(C \cdot N \cdot D)$—linear in context length, compared to $\mathcal{O}(N^2 D)$ for exact attention—and produces $(N/B) \times C_q \times C_k$ summaries (Steps 1–2 in Figure 1).

**Causal accumulation**  To enable causal attention, we accumulate summaries along the spatial axis so that block $b$ has access to summaries of all preceding blocks $0, \dots, b-1$. The accumulation is *exclusive*—a block does not include itself:

- $\overline{\mu}$: cumulative logsumexp
- $\overline{k}, \overline{v}$: cumulative softmax-weighted sum (weights derived from accumulated $\overline{\mu}$)

After accumulation, accumulated$_{ijb}$ summarizes all keys/values in blocks $0 \dots b-1$ belonging to key cluster $j$, specialized for query cluster $i$ via exponential tilting (Step 3 in Figure 1).

We implement this sequentially rather than with parallel scan (prefix sum), since there is already massive parallelism from $C_q \times C_k \times$ heads $\times$ batch independent accumulations.

### 3.2. Two-Level Retrieval

The monopole approximation in Eq. (3) drops the residual cross term $\tilde{q} \cdot \tilde{k}$. This error is significant only when the residual query has large positive dot product with keys in a cluster. Retrieval selects exactly these clusters for exact token-level attention; for the remaining clusters, keys either carry negligible attention weight or have small residual interaction, so the monopole approximation holds. Intuitively, if the attention pattern is smooth, weighted centroids approximate it well; if it is sharp, the high-weight keys are few and retrieval captures them. This decomposition—approximating smooth interactions, computing sharp ones exactly—mirrors the Fast Multipole Method (Rokhlin, 1985; Greengard & Rokhlin, 1987), with relevance in representation space playing the role of spatial distance. This proceeds in two stages: first selecting *clusters*, then selecting *spatial blocks* within those clusters.

**Cluster selection**  For each query $q = \overline{q}_i + \tilde{q}$ in spatial block $b$, we compute attention scores to all $C_k$ accumulated

cluster summaries:

$$\text{score}_j = \tilde{q} \cdot \overline{k}_{ijb} + \overline{\mu}_{ijb} \qquad (7)$$

This is the best available estimate of the log attention mass that query $q$ will receive from cluster $j$. We select the top-$k_1$ clusters for exact retrieval (e.g., $k_1 = 8$ out of $C = 128$), mask them out pre-softmax, and compute approximate attention over the remaining $C - k_1$ clusters using the scores of Eq. (7) and value summaries of Eq. (5) (Step 4 in Figure 1).

**Spatial block selection**  For queries that selected cluster $j$, we now select which spatial blocks within cluster $j$ to retrieve. This uses the *pre-accumulation* per-block summaries rather than the accumulated ones.

Queries are segmented by which cluster(s) they are retrieving. For queries in query cluster $i$ that selected key cluster $j$, we attend to the $N/B$ per-block summaries for the $(i, j)$ pair, apply a block-index causal mask ($b_q > b_k$), and select the top-$k_2$ spatial blocks (e.g., $k_2 = 1$ out of $N/B = 8$). Selected blocks are masked out pre-softmax, and we compute approximate attention over the remaining blocks as above, but with *pre-accumulation* summaries (Step 5 in Figure 1).

The output specifies $k_1 \times k_2$ (cluster, block) pairs per query for exact retrieval (e.g., $8 \times 1 = 8$ pairs).

**Dipole corrections**  An alternative to retrieval is to improve the monopole (centroid) approximation with higher-order terms. By expanding attention as a polynomial in $\tilde{q}$, one obtains a dipole correction involving the covariance $\text{Cov}(v, k)$ within each cluster, reducing approximation error from $\mathcal{O}(\tilde{q} \cdot \tilde{k})$ to $\mathcal{O}(\tilde{q}^2 \cdot \tilde{k}^2)$. We derive this in Appendix B. In practice, retrieval provides larger accuracy gains at acceptable computational cost (see Appendix B for an empirical comparison), so we use retrieval in our main experiments.

### 3.3. Exact Retrieval and Local Attention

The retrieved (cluster, block) pairs and the local block diagonal are computed with exact flash attention, recovering token-level detail that the monopole summaries cannot capture.

**Exact retrieval**  We perform exact flash attention from each query to its selected (cluster, block) pairs, using the *full* query (not the residual) (Step 6 in Figure 1). Keys and values are bucketed by (cluster, block). We build an inverse index mapping each (cluster, block) to the queries that retrieve it, then perform segmented flash attention.

To avoid $k_1 \times k_2$ memory blowup, we use index arrays into the original queries, keys, and values rather than materializing expanded arrays, similar to the segmentation approach in FlashMoBA (Xiao et al., 2025).

**Local attention** For within-block attention (the "diagonal"), we use exact flash attention with standard causal masking. This handles local interactions where attention is typically strongest. (Figure 1 depicts only the far-field approximation; local attention is computed separately.)

**Output merging** The final output merges four components via logsumexp weighting:

1. Approximate attention from non-retrieved clusters (from cluster selection)

2. Approximate attention from non-retrieved blocks within retrieved clusters (from block selection)

3. Exact attention from retrieved (cluster, block) pairs

4. Exact local attention within the current block

**Complexity** At our operating point (64k context, 8k block diagonal), the block diagonal contributes 1/8 of full quadratic cost, with the far-field approximated at $64\times$ sparsity. Maintaining these ratios as context scales, we find approximation error decreases substantially (Appendix F.6); quantifying the resulting speedups at longer contexts is future work. With typical parameters ($N = 2^{16}$, $C = 128$, $B = 8192$, $k_1 = 8$, $k_2 = 1$), we achieve significant speedups both in theoretical FLOPs and in wall-clock time (Section 4). We implement MuSe in JAX with custom Pallas kernels for summary computation and segmented retrieval, and CUDA kernels for K-means clustering (Appendix C). The approximation is compatible with standard distributed training strategies (Appendix H).

## 4. Experiments

### 4.1. Setup and Microbenchmarks

**Runtime Comparison** Table 1 compares attention runtime across methods on the 1B model (64 heads, 2 sequences of 64k context each). MuSe achieves $2\times$ speedup over CUDNN Flash Attention while maintaining high approximation quality. We also compare against MoBA at matched far-field sparsity: both methods use an 8k block diagonal, and MoBA retrieves one 512-token block per query to match MuSe's retrieval of 8 clusters $\times$ 1 spatial block ($\sim$512 tokens). MoBA is slightly faster than MuSe (no summarization overhead), but MuSe achieves $40\times$ lower approximation error (0.006 vs 0.248 relative squared error).

**1B Model Validation** Table 2 validates the query clustering benefit on our headline 1B parameter model (320 heads, head dimension 64). We vary cluster count jointly (Q=K) with retrieval fraction fixed at R/QK = 1/16. For each configuration, we compute the effective no-query-clustering

*Table 1.* **MuSe achieves $2\times$ speedup with $40\times$ lower error than MoBA at matched sparsity.** Attention runtime comparison (1B model, 64 heads, $2\times$64k context). [†]Matched sparsity: 8k block diagonal, $64\times$ far-field.

| Method | Time (ms) | Speedup | RSE |
|---|---|---|---|
| CUDNN Flash | 225.6 | $1.00\times$ | 0 (exact) |
| Pallas Flash | 265.6 | $0.85\times$ | 0 (exact) |
| MuSe (Ours)[†] | 114.1 | $1.98\times$ | **0.006** |
| MoBA[†] | 109.0 | $2.07\times$ | 0.248 |
| Block diag. only | 35.0 | $6.45\times$ | 0.481 |

cluster count—the cluster count the ablated method would require to achieve the same error, interpolated via power-law fit. MuSe exhibits a steeper power-law slope ($d \approx -0.89$) than the no-query-clustering ablation ($d \approx -0.59$), so the effective cluster multiplier grows with cluster count: from $5.1\times$ at QK16 to $10.7\times$ at QK512. At our operating point of QK128R8, MuSe achieves error that would require $\sim$1200 clusters without query clustering—a $9.2\times$ effective cluster count advantage.

*Table 2.* **Query clustering provides a $\sim 9\times$ effective cluster count advantage.** MuSe approximation quality on 1B model (320 heads, 2 sequences, 64k context). R = QK/16 throughout. Timing measured on 40 heads due to memory constraints. Effective No-Q cluster count computed via power-law interpolation; values marked with † are extrapolated beyond measured No-Q data (QK512).

| QK | RSE | Corr | Eff. No-Q | Mult. | Time (ms) |
|---|---|---|---|---|---|
| 16 | 0.03797 | 0.9775 | 82 | $5.1\times$ | — |
| 32 | 0.02100 | 0.9878 | 191 | $6.0\times$ | 130.0 |
| 64 | 0.01139 | 0.9935 | 568† | $8.9\times$ | 88.5 |
| **128** | **0.00622** | **0.9965** | **1179†** | **$9.2\times$** | **80.5** |
| 256 | 0.00330 | 0.9980 | 2518† | $9.8\times$ | 106.8 |
| 512 | 0.00173 | 0.9990 | 5481† | $10.7\times$ | 219.8 |

**Monopole vs Retrieval Decomposition** To understand where query clustering helps, we measure approximation quality for monopole-only (no retrieval) and retrieval-only (no monopole) variants on the 1B model. Table 3 shows that query clustering dramatically improves monopole quality ($3.7$–$5.6\times$ error reduction) but barely affects retrieval selection ($1.02\times$). This confirms that query clustering improves the *quality* of cluster summaries; retrieval selection works well either way because it only needs to identify high-attention clusters, not compute precise values.

### 4.2. Pretraining Results

**Headline Results** Table 4 summarizes our main 1B parameter results on both code and scientific PDF domains. On code, MuSe achieves a train-MuSe/test-MuSe loss of 0.7001 compared to the CUDNN baseline of 0.7026 (0.4% improvement). When tested with CUDNN attention, there is a small adaptation gap (0.7108, 1.2% degradation); however, this

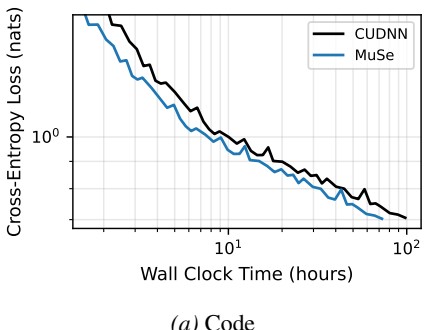

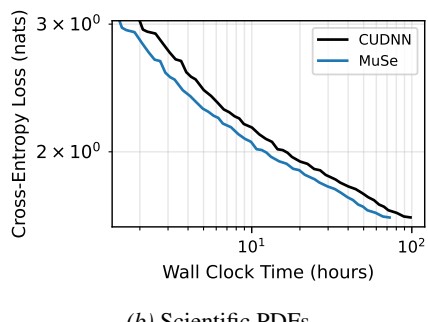

*(a)* Code

*(b)* Scientific PDFs

*Figure 3.* Training loss versus wall-clock time for 1B models on code (left) and scientific PDF (right) domains. MuSe (blue) and CUDNN (black) follow the same loss trajectory, shifted horizontally—the gap is the 36% throughput advantage. Plots start after warmup (4000 steps). Trained on a single node of 8 NVIDIA A100 GPUs.

*Table 3.* **Query clustering acts through improved cluster summaries, not retrieval selection.** Monopole vs retrieval decomposition on 1B model. "With Q" uses query clustering; "No Q" uses zeroed query centroids. Query clustering provides $4.5\times$ benefit for monopole but only $1.02\times$ for retrieval.

| Component | Setting | RSE | Corr | Q benefit |
|---|---|---|---|---|
| Monopole (K64) | With Q | 0.0342 | 0.9795 | $3.7\times$ |
|  | No Q | 0.1278 | 0.9282 |  |
| Monopole (K128) | With Q | 0.0227 | 0.9866 | $4.5\times$ |
|  | No Q | 0.1016 | 0.9426 |  |
| Monopole (K256) | With Q | 0.0148 | 0.9913 | $5.6\times$ |
|  | No Q | 0.0824 | 0.9534 |  |
| Retrieval (K128R8) | With Q | 0.2429 | 0.9407 | $1.02\times$ |
|  | No Q | 0.2475 | 0.9367 |  |

*Table 4.* **MuSe matches or beats the CUDNN baseline on both domains.** Headline 1B results on code (24B tokens) and scientific PDFs (24B tokens). Cross-entropy loss, lower is better. [†]Fine-tuned with CUDNN attention for 0.1% of pretraining tokens.

| Domain | Train | Test Attention | |
|---|---|---|---|
|  |  | CUDNN | MuSe |
| Code | CUDNN | 0.7026 | — |
| Code | MuSe | 0.7108 | 0.7001 |
| Code | MuSe[†] | **0.6994** | 0.7020 |
| PDF | CUDNN | 1.6201 | — |
| PDF | MuSe | **1.6166** | 1.6188 |

**Speedup Analysis**  At 1B scale on a single node of 8 A100 GPUs with 64k context, MuSe achieves 88.9 KTok/s compared to 65.3 KTok/s for CUDNN Flash Attention—a **36% throughput improvement**. Our Pallas kernels are not highly optimized relative to production-quality CUDA; substantially larger speedups are achievable given the $64\times$ far-field sparsity. Combined with the minimal quality degradation shown above, this demonstrates that MuSe provides substantial practical speedups for long-context pretraining.

Figure 3 shows training loss versus wall-clock time on both code and scientific PDF domains. At any given loss level, MuSe reaches that point faster than CUDNN, with the horizontal gap representing the throughput advantage. The curves are nearly identical when plotted against tokens (see Appendix), confirming that MuSe achieves equivalent sample efficiency—the speedup comes purely from faster iteration, not from any change in learning dynamics.

**Where the Time Goes**  Table 5 breaks attention runtime down by component at our 1B operating point (QK128R8, 64k context). Retrieval and the block-diagonal exact attention together account for two-thirds of the cost, while monopole summarization and clustering, the two components central to our novel use of query-cluster-specialized exponential tilting, are minor contributors at 8.8% and 8.4%.

gap closes rapidly with brief CUDNN fine-tuning: after just 26M tokens ($\sim$0.1% of pretraining), the fine-tuned model achieves 0.6994, *beating* the baseline (see Appendix G.5 for details). We further identify the gradient pathway responsible for this adaptation, namely gradients through the tilted aggregation weights of the monopole summary, and preliminarily find that making it non-differentiable during training removes the gap with no fine-tuning at all, reaching 0.6991 under exact attention (Appendix C, Table 11).[2] On scientific PDFs, MuSe outperforms the baseline: 1.6166 vs 1.6201 (0.2% improvement), with minimal interchangeability gap (1.6188 when tested with MuSe attention). We hypothesize two competing effects: the approximation acts as implicit regularization (improving generalization), while extended training allows minor adaptation to the approximation's specific behavior. The regularization effect dominates early and at smaller scales, explaining why MuSe beats the baseline; adaptation emerges with extended training but is easily removed via fine-tuning.

---

[2]Identified during review of this work; our main results retain the fine-tuning workflow.

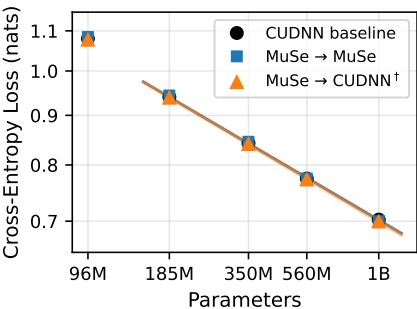

*Figure 4.* Scaling behavior from 96M to 1B parameters. Power law fits (lines) span 185M–1B, where scaling is cleanest; the 96M point falls slightly above the fit. MuSe-trained models evaluated with CUDNN attention (orange triangles, †) track the exact attention baseline, confirming that the approximation preserves scaling properties. †The 1B point uses the fine-tuned value after 0.1% additional CUDNN training.

*Table 5.* Attention runtime by component at the 1B operating point (QK128R8, 64k context, head dimension 64). Retrieval dominates; monopole summarization and clustering, central to MuSe's query-cluster-specialized tilting, together account for only ∼17%.

| Component | Share of runtime |
|---|---|
| Retrieval | 37% |
| Block-diagonal exact attention | 30% |
| XLA fusions, sorts, transposes | 18% |
| Monopole summarization | 8.8% |
| Clustering | 8.4% |

The dominant cost is therefore retrieval rather than the approximation machinery itself, making retrieval the natural target for further kernel optimization. Clustering shares the $O(NCD)$ scaling of the monopole stage (Appendix C), so its share stays roughly constant as cluster count grows and it cannot become a bottleneck.

**Scaling Analysis** Table 6 and Figure 4 present scaling results on the code domain from 96M to 1B parameters. We train each model for approximately 20 tokens per parameter, following compute-optimal scaling (Hoffmann et al., 2022). We report cross-entropy loss for models trained with either CUDNN Flash Attention or MuSe, evaluated with both attention methods to assess interchangeability. At scales up to 560M, MuSe-trained models match or exceed the CUDNN baseline when evaluated with CUDNN attention. At 1B scale, MuSe-trained models tested with MuSe attention beat the baseline (0.7001 vs 0.7026), though there is minor adaptation when tested with CUDNN attention (0.7108); this adaptation is removed with brief CUDNN fine-tuning (0.1% of pretraining tokens), after which the model beats the baseline even when tested with CUDNN attention (0.6994 vs 0.7026). Power law fits to the 185M–1B data (excluding the undertrained 96M point) show that MuSe follows the same scaling law as exact attention (loss $\propto$ params$^{-0.17}$,

*Table 6.* **MuSe matches baseline scaling laws across five model sizes.** Scaling results on code domain (cross-entropy loss). Lower is better. †Fine-tuned with CUDNN attention for 0.1% of pretraining tokens.

| Params | Train | Test Attention | |
|---|---|---|---|
| | | CUDNN | MuSe |
| 1B | CUDNN | 0.7026 | — |
| 1B | MuSe | 0.7108 | 0.7001 |
| 1B | MuSe† | **0.6994** | 0.7020 |
| 560M | CUDNN | 0.7745 | — |
| 560M | MuSe | **0.7728** | 0.7746 |
| 350M | CUDNN | 0.8427 | — |
| 350M | MuSe | **0.8407** | 0.8453 |
| 185M | CUDNN | 0.9400 | — |
| 185M | MuSe | **0.9384** | 0.9435 |
| 96M | CUDNN | 1.080 | — |
| 96M | MuSe | **1.077** | 1.084 |

*Table 7.* **MuSe beats the exact baseline; MoBA does not.** Method comparison at 185M and 560M scale (cross-entropy loss). All methods use matched sparsity (8k block diagonal, 64× far-field sparsity). Lower is better.

| Method | Test Attention | |
|---|---|---|
| | CUDNN | Method |
| *185M:* | | |
| CUDNN (baseline) | 0.9400 | — |
| MuSe | **0.9384** | 0.9435 |
| MoBA | 0.9714 | 1.0027 |
| *560M:* | | |
| CUDNN (baseline) | 0.7745 | — |
| MuSe | **0.7728** | 0.7746 |
| MoBA | 0.7958 | 0.8209 |

$R^2 > 0.999$).

**Method Comparison** Table 7 compares MuSe against MoBA at 185M and 560M scale with matched sparsity (8k block diagonal, 64× far-field sparsity). MoBA uses spatial block structure without semantic clustering, retrieving one 512-token block in the far field per query to achieve the matched sparsity.

MuSe is the only method that *beats* the exact CUDNN baseline at both scales (0.9384 vs 0.9400 at 185M; 0.7728 vs 0.7745 at 560M), demonstrating that our approximation can act as a beneficial regularizer. MoBA performs significantly worse at both scales (0.9714 at 185M, 0.7958 at 560M), showing that semantic clustering—not just sparsity—is critical for approximation quality. Even if MoBA reallocates compute from the block diagonal to reduce far-field sparsity to 8×, MuSe maintains an advantage (Appendix F.5). The importance of query clustering is discussed further in Section 4.3.

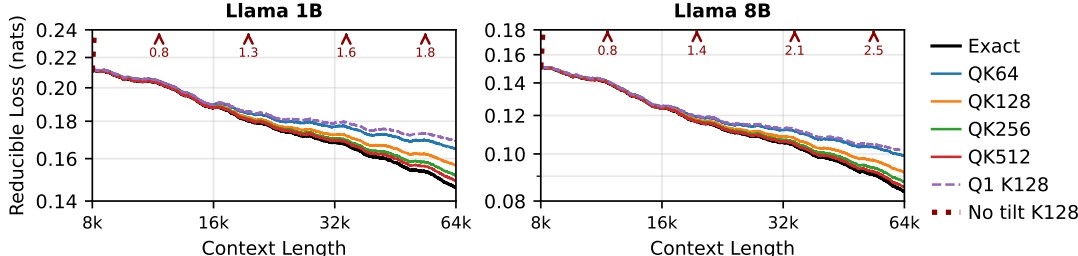

*Figure 5.* Cumulative mean loss versus context length on Project Gutenberg text for Llama 3.2-1B (left) and Llama 3.1-8B (right). MuSe with various cluster counts (QK64–QK512) tracks exact attention closely, with the gap reducing by $\sim 1.8\times$ per doubling for 1B and $\sim 2\times$ for 8B. Without exponential tilting ("No tilt K128", dotted), loss explodes immediately. Query clustering provides additional benefit: QK64 outperforms Q1 K128 despite using half the key clusters, demonstrating that clustering queries more than doubles practical quality. Y-axes show reducible loss (cumulative mean minus fitted irreducible loss: 2.30 nats for 1B, 1.96 nats for 8B).

**Head Dimension** Following recent open-weight releases from major labs (OpenAI, 2025), our main experiments use head dimension 64. Table 8 validates that MuSe also works well with head dimension 128, as used by Llama models. At 185M scale, MuSe with head dimension 128 slightly outperforms the CUDNN baseline (0.9113 vs 0.9121) and shows strong interchangeability. Microbenchmarks confirm similar approximation quality: at our operating point (QK128R8), head dimension 128 achieves 0.0100 relative squared error and 0.994 correlation, compared to 0.0094 and 0.995 for head dimension 64.

*Table 8.* **MuSe generalizes to head dimension 128.** Head dimension 128 comparison at 185M scale (cross-entropy loss).

| Train | Test Attention | |
|---|---|---|
| | CUDNN | MuSe |
| CUDNN | 0.9121 | 0.9267 |
| MuSe | **0.9113** | 0.9159 |

### 4.3. Generalization and Long-Context Validation

**Evaluation on Existing Models** To validate that MuSe generalizes beyond models trained with it, we evaluate on pretrained Llama 3.2-1B (head dimension 64) and Llama 3.1-8B (head dimension 128) using 64k-token passages from Project Gutenberg. Figure 5 shows cumulative mean loss as a function of context length. With tilting enabled, all MuSe configurations track exact attention with modest gaps that decrease with cluster count. The gap approximately halves with each doubling of QK clusters, and QK512 nearly matches exact attention for both models.

Critically, without exponential tilting by query centroids ("No tilt K128"), loss explodes immediately—the approximation fails catastrophically on pretrained models. This demonstrates that tilting is essential for drop-in compatibility: attention heads with non-trivial mean query values cannot be approximated without accounting for this bias.

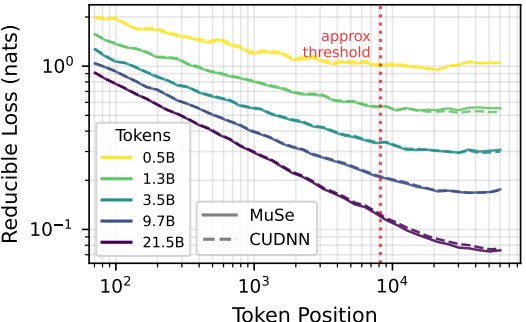

*Figure 6.* Loss versus position during 1B model training on code data. Curves show progression from early training (light) to late training (dark) at 5 checkpoints spanning 0.5B to 21.5B tokens. MuSe (solid) closely tracks cuDNN attention (dashed) throughout training. The vertical line marks $2^{13} = 8192$ tokens, beyond which MuSe uses far-field approximation. Both methods learn to utilize the full 64k context, with loss continuing to decrease even in the approximated region.

Query clustering provides substantial additional benefit beyond tilting alone. Comparing Q1 K128 (single query cluster with tilting) to QK64 (64 query and key clusters), QK64 achieves lower error despite using half as many key clusters. This shows that clustering queries more than doubles the effective quality of the approximation. This benefit comes at negligible cost: attention with few queries is memory-bound, with operational intensity approximately proportional to the query count (Ye et al., 2025), so tilting by moderate numbers of centroids (32–64) is essentially free on current hardware.

**Context Utilization During Training** Figure 6 demonstrates that models trained with MuSe learn to utilize the full 64k context throughout training. Loss continues to decrease with position even beyond the approximation threshold at $2^{13}$ tokens, where MuSe transitions from exact to far-field computation. The close correspondence between MuSe and exact attention at all training stages confirms that the approximation does not impair the model's ability to learn

long-range dependencies.

**Long-Context Retrieval Evaluation** A commonly used proxy for long-context downstream capability is retrieval, probed by standardized evaluations such as RULER (Hsieh et al., 2024). Since MuSe is used only during training and replaced by exact attention at inference, such evaluations do not test MuSe attention directly; they test whether a model trained with MuSe has learned to use the retrieval capability of exact attention once it is restored. A strong result is therefore a further demonstration of transferability between MuSe and exact attention, compounding the pretrained-model (Llama) generalization shown above. RULER, however, targets powerful generalist instruction-tuned models rather than the 1B-scale, domain-specific base models we train. Both MuSe- and CUDNN-trained models therefore score poorly on most RULER tasks, the code models in particular near zero on everything but basic single-needle retrieval (needle-in-a-haystack, NIAH); we read this as reflecting domain shift and limited instruction-following far more than a retrieval deficit. Per-task results are reported in Table 31.

Rather than attempt to disentangle the contributions of this domain shift and limited instruction-following capability to the complex-variant RULER scores, we designed a custom NIAH evaluation that is in-domain and natural for our code models, a domain where retrieval is intrinsic: function definitions are retrieved via imports across long distances, variable bindings span large scopes, and cross-file dependencies are pervasive. It uses cross-file function imports as the retrieval primitive: a file defining a function (the *key*) is placed in a repository, and a separate file imports and calls it (the *query*); the function name carries a random uid (the *value*), so correct completion requires retrieval rather than guessing. We construct single-needle, multi-key ($K \in \{2, 4\}$ distractor definitions), and multi-query ($N \in \{2, 4\}$ needles probed at different depths) variants. Table 9 reports uid-level exact match aggregated over needle depth: MuSe-trained models match or exceed exact-attention-trained models on every variant. Resolving by depth (Appendix G.4), MuSe-trained accuracy remains above 89% in every depth bucket, whereas the exact-attention baseline degrades with depth on the harder variants (e.g., 64–97% across depth buckets on multi-key $K$=4).

**Downstream Evaluation** As a holistic, real-task long-context benchmark we evaluate the code-completion subset of LongBench (Bai et al., 2024) (RepoBench-P and LCC), which is in-distribution for our code models; the remaining LongBench tasks require instruction following. MuSe matches exact attention on both subtasks (Table 10), with differences comparable to the reported standard error.

We additionally evaluate 1B models on ARC-Easy and

*Table 9.* **MuSe-trained models match or exceed exact-attention-trained models on in-distribution retrieval.** Custom code NIAH at 64k context: uid-level exact match ($\pm$ binomial std. error). Column headers denote the training regime; all evaluation uses exact attention.

| Task | $n$ | CUDNN | MuSe |
|---|---|---|---|
| Single-needle | 2547 | $97.80 \pm 0.29\%$ | $\mathbf{98.35 \pm 0.25\%}$ |
| Multi-key 2× | 2475 | $92.73 \pm 0.52\%$ | $\mathbf{98.46 \pm 0.25\%}$ |
| Multi-key 4× | 2421 | $81.50 \pm 0.79\%$ | $\mathbf{97.36 \pm 0.33\%}$ |
| Multi-query 2× | 538 | $88.48 \pm 1.38\%$ | $\mathbf{98.70 \pm 0.49\%}$ |
| Multi-query 4× | 1060 | $83.58 \pm 1.14\%$ | $\mathbf{98.77 \pm 0.34\%}$ |

*Table 10.* **MuSe matches exact attention on LongBench code completion.** Edit similarity ($\pm$ harness-reported std. error). Both methods are within the reported standard error of each other.

| Task | CUDNN | MuSe |
|---|---|---|
| RepoBench-P | $0.382 \pm 0.012$ | $0.393 \pm 0.011$ |
| LCC | $0.329 \pm 0.009$ | $0.314 \pm 0.010$ |

SciQ with the lm-eval harness. MuSe-trained and CUDNN-trained models perform comparably (e.g., 48.7% vs 47.9% on ARC-Easy), with all differences within one standard error, confirming that the approximation does not impair downstream task performance (full results in Appendix G.2).

## 5. Conclusion

We have presented Multipole Semantic Attention (MuSe), an efficient approximation to softmax attention that clusters queries and keys separately in their learned representation spaces. By computing query-specific cluster summaries and retrieving high-attention clusters, MuSe achieves high far-field sparsity with under 1% relative squared error.

Our experiments demonstrate practical benefits at scale: 2× speedup over CUDNN Flash Attention on isolated attention layers, and 36% wallclock throughput improvement when pretraining 1B parameter models at 64k context. Models trained with MuSe achieve comparable loss to exact attention baselines and remain interchangeable at test time. Exponential tilting by query centroids is essential for compatibility with pretrained models, and query clustering provides substantial additional quality at negligible cost.

Limitations include the focus on head dimension 64 in most experiments, though preliminary results at dimension 128 are promising. Future work includes scaling to larger models, kernel optimization, and investigating whether query-dependent tilting benefits other approximate attention mechanisms. MuSe applies to grouped-query attention without modification, as exact attention does; since the keys are shared within a group, clustering them once per group rather than once per query head is a natural further optimization.

# Acknowledgements

This work was supported by the Hessian research priority programme LOEWE within the project "WhiteBox", and the Aleph Alpha Collaboration Lab 1141. It benefited from the Federal Ministry for Research, Technology and Space (BMFTR) project "XEI: Extremely Efficient Inference for Large Context Length" (XEI), project identification number 01IS24079B, and from funding by the Deutsche Forschungsgemeinschaft (DFG, German Research Foundation) under Germany's Excellence Strategy – EXC-3057. We thank Manuel Brack, Moritz Willig, Felix Friedrich, Felix Divo, Florian Busch, and Rubin Härle for helpful discussions and feedback on the manuscript.

# Impact Statement

Our work on Multipole Semantic Attention (MuSe) has significant potential for democratizing access to long-context language models by substantially reducing computational costs. This could enable broader participation from resource-limited researchers and reduce environmental impact, though network effects may also happen. The increased efficiency in processing longer contexts may benefit applications in scientific research, education, and document analysis. However, these same capabilities raise concerns about potential misuse in automated disinformation campaigns, enhanced surveillance through efficient processing of large text corpora, and displacement of knowledge workers. The interchangeability of MuSe-trained models with standard attention at inference time may complicate accountability and bias auditing efforts. There is also the risk of over-emphasizing computational efficiency at the expense of other crucial aspects like robustness, factual accuracy, and alignment, necessitating careful monitoring of downstream applications and continued research into behavioral differences emerging from approximate attention mechanisms.

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

# A. Algorithm Pseudocode

---

**Algorithm 1** MuSe Causal Attention

---

**Require:** Queries $Q$, Keys $K$, Values $V$ of length $N$; block size $B$; cluster count $C$; retrieval counts $k_1$, $k_2$

**Ensure:** Output $O$ of length $N$

1: **// Clustering (global, ignores block boundaries)**
2: Cluster $Q$ into $C$ clusters; decompose $q = \overline{q}_i + \tilde{q}$
3: Cluster $K$ into $C$ clusters; decompose $k = \overline{k}_j + \tilde{k}$
4: **// Per-block summaries:** $C \times C \times (N/B)$ **entries**
5: **for** each spatial block $b$, query cluster $i$, key cluster $j$ **do**
6: $\quad \overline{\mu}_{ijb}, \overline{k}_{ijb}, \overline{v}_{ijb} \leftarrow$ attention-weighted stats of $(K,V)$ in $(j,b)$ from $\overline{q}_i$
7: **end for**
8: **// Causal accumulation along spatial axis**
9: **for** each block $b = 1, \ldots, N/B - 1$ **do**
10: $\quad$ accum$_{ij,b} \leftarrow$ accum$_{ij,b-1} \oplus$ summary$_{ij,b-1}$ {logsumexp merge}
11: **end for**
12: **// Two-level retrieval + attention (per query)**
13: **for** each query $q$ in cluster $i$, block $b$ **do**
14: $\quad$ Select top-$k_1$ key clusters from accum$_{i,:,b}$ {Cluster selection}
15: $\quad$ Select top-$k_2$ blocks per cluster from summary$_{i,j,:}$ {Spatial selection}
16: $\quad O_{\text{retrieved}} \leftarrow$ exact attention on $k_1 \times k_2$ (cluster, block) pairs
17: $\quad O_{\text{approx}} \leftarrow$ approximate attention on remaining clusters/blocks
18: $\quad O_{\text{local}} \leftarrow$ exact causal attention within block $b$
19: **end for**
20: **return** logsumexp-weighted merge of $O_{\text{retrieved}}$, $O_{\text{approx}}$, $O_{\text{local}}$

---

# B. Dipole Derivation

Here we present an alternative to retrieval based on polynomial expansion of the attention function, which we call the dipole correction. While retrieval performs better empirically, this derivation provides theoretical insight into the approximation.

**Polynomial Expansion via Cumulant Generating Functions** The keys $k$ of some cluster $j$ can be considered as a probability distribution of equally weighted point masses. They have moment generating function $\mathcal{M}_j(q) := \mathbb{E}_{k \in C_j} \exp(q \cdot k)$ and cumulant generating function (CGF) $\mathcal{K}_j(q) := \ln \mathcal{M}_j(q)$, whose derivatives give the cumulants (mean, variance, skewness, etc.).

Attending to keys with query centroid $\overline{q}_i$ corresponds to an exponential tilt of this distribution. Define the CGF of the joint key-value distribution: $\mathcal{K}_j(q,t) := \ln \mathbb{E}_{k,v \in C_j} \exp(q \cdot$

$k + t \cdot v)$. The output of softmax attention on cluster $j$ for query $\overline{q}$ is: $V_j(\overline{q}) = \left. \frac{\partial \mathcal{K}_j(q,t)}{\partial t} \right|_{q=\overline{q}, t=0}$.

For a query $q = \overline{q}_i + \tilde{q}$, we can approximate $V_j(q)$ by polynomial expansion around $\overline{q}_i$:

$$V_{ij}(\tilde{q}) = \overline{v}_{ij} + \text{Cov}_{ij}(v,k)\tilde{q} + \frac{1}{2}\tilde{q}^T \text{Skew}_{ij}(v,k,k)\tilde{q} + \ldots \tag{8}$$

where $\overline{v}_{ij}$, $\text{Cov}_{ij}(v,k)$, and $\text{Skew}_{ij}(v,k,k)$ are cumulants of the exponentially-tilted distribution.

Similarly, the unnormalized attention weight expands as:

$$M_{ij}(\tilde{q}) = M_j(\overline{q}_i) \exp\left( \tilde{q} \cdot \overline{k}_{ij} + \frac{1}{2}\tilde{q}^T \text{Cov}_{ij}(k,k)\tilde{q} + \ldots \right) \tag{9}$$

**Dipole Truncation** Retaining only terms linear in $\tilde{q}$ gives the dipole approximation with error $\mathcal{O}(\tilde{q}^2\tilde{k}^2)$. The covariance matrices $\text{Cov}_{ij}(v,k)$ can be pre-merged across key clusters for each query cluster, reducing complexity from $\mathcal{O}(NCD^2)$ to $\mathcal{O}(ND^2)$. The expected error is $\mathcal{O}(\text{Tr}(\text{Cov}(\tilde{q}, \tilde{q})\,\text{Cov}(\tilde{k},\tilde{k})))$—exactly the quantity minimized by K-means, motivating our clustering choice.

**Partially Specialized Dipoles** The above error bounds apply to fully specialized dipoles $\text{Cov}_{ij}(v,k)$, which require $\mathcal{O}(QND^2)$ compute to construct (where $Q$ is the number of query clusters and $N$ is the sequence length). To reduce this cost, we instead compute a single $\text{Cov}_j(v,k)$ for each key cluster without the exponential tilt by $\overline{q}_i$, reducing construction cost to $\mathcal{O}(ND^2)$. This degrades the error bound from $\mathcal{O}(\tilde{q}^2\tilde{k}^2)$ to $\mathcal{O}(\overline{q} \cdot \tilde{q} \cdot \tilde{k}^2)$, introducing dependence on the query centroid magnitude. This degradation is substantial—roughly halving the error reduction relative to monopole-only—but acceptable in exchange for the dramatic reduction in complexity.

**Compatibility with Retrieval** One might hope to combine dipole corrections with retrieval for further accuracy gains. However, the natural dipole aggregation produces a single correction per query cluster by weighting each $\text{Cov}_j(v,k)$ according to how much that query cluster attends to each key cluster. When retrieval then selects specific key clusters for exact attention computation, those clusters have already contributed to the aggregated dipole term. Correctly combining the two would require subtracting out the retrieved clusters' contribution from the dipole before adding the exact attention—possible, but complex. Given that retrieval alone already provides larger accuracy gains than dipoles, we opted not to pursue this combination.

**Empirical Comparison with Retrieval** In practice, we found that retrieval of high-attention clusters provides bet-

ter accuracy than dipole corrections. At K128, monopole-only approximation achieves 0.0227 relative squared error; adding dipole corrections reduces this to 0.0150 (1.5× improvement), while adding retrieval instead reduces it to 0.0062 (3.6× improvement). We therefore use retrieval in our main experiments.

## C. Clustering Details

We use streaming K-means to obtain cluster centroids, followed by a final assignment pass.

**Centroid Computation**    Centroids are computed globally across the entire sequence. Given $N$ input vectors (queries or keys), we compute $K$ cluster centroids as follows:

1. **Initialization:** Shuffle inputs with a fixed random seed, then uniformly subsample every $\lfloor N/K \rfloor$-th vector to obtain $K$ initial centroids.

2. **Streaming update:** Process all $N$ vectors in a single pass using minibatches of 64 vectors. For each minibatch, compute similarities to current centroids using tensor cores, then assign each vector to its nearest centroid. For each assigned vector $x$, update the centroid total and count as $t \leftarrow \beta \cdot t + x$ and $c \leftarrow \beta \cdot c + 1$ (with $\beta = 0.9$), processing assignments sequentially within the minibatch. The centroid is recovered as $t/c$.

A single pass suffices because the effective sample size per cluster grows with sequence length; at 64k tokens with 128 clusters, each cluster sees ∼500 vectors on average. All clustering computations use float16 precision for efficiency.

**Balanced Assignment**    After computing global centroids, assignment is performed independently within each spatial block. Each vector is assigned to its nearest centroid subject to a maximum cluster size of $4 \times B/K$ per block, where $B$ is the spatial block size. This static cap simplifies prototyping with pure JAX implementations; the final Pallas kernels could handle variable cluster sizes but we retain the cap for consistency. The 4× average factor was chosen empirically as it rarely binds in practice. Squared distances are computed efficiently via the identity $\|x - c\|^2 = \|x\|^2 - 2\langle x, c \rangle + \|c\|^2$, where $\|x\|^2$ terms cancel when comparing distances to different centroids.

**Centroid Gradient Flow**    When computing attention from query centroids (for exponential tilting), we stop gradients from flowing back through the centroid to its constituent queries. This prevents queries from being trained to "cluster well," a property that provides no benefit when switching to exact attention at test time. We found this reduces adaptation effects.

**Aggregation Weight Gradient Flow**    The initial summary step aggregates keys and values using softmax weights tilted by the query centroid $\overline{q}$. As an optional alternative to the brief fine-tuning of Appendix G.5, we treat these tilted aggregation weights as non-differentiable with respect to both $\overline{q}$ and the keys. This does not destroy gradient signal to the keys: they are still trained through the derivative of the loss with respect to the *aggregated* keys (the key centroids that enter the summary). The removed term is specific to *tilted* aggregation of keys/values, via the dependence of that tilting effect on the keys themselves. Empirically, stopping this gradient slows learning slightly early in training but converges to better non-adapted minima in larger-scale runs: on the 1B code model it removes the adaptation gap entirely, beating the baseline when evaluated with exact attention and without any fine-tuning (Table 11).

*Table 11.*  Effect of treating the tilted aggregation weights as non-differentiable, on the 1B code model evaluated with exact (CUDNN) attention. The non-differentiable-aggregation variant removes the adaptation gap with no fine-tuning.

| Model (evaluated with CUDNN attention) | Loss |
|---|---|
| CUDNN baseline | 0.7026 |
| MuSe, default aggregation | 0.7108 |
| MuSe, default aggregation + fine-tune | 0.6994 |
| MuSe, non-differentiable aggregation | **0.6991** |

**Clustering Cost**    The cost of clustering is dominated by computing the similarity of all $N$ tokens against the $C$ centroids, a matmul of size $N \times C \times D$. Clustering performs four such matmuls, compared with five in the monopole forward pass and twelve in its backward pass, so clustering is under 20% of the aforementioned computation alone, and a smaller fraction still once retrieval and block-diagonal attention are counted (Table 5). Both clustering and the monopole stage scale as $O(NCD)$, strictly cheaper than the $O(N^2D)$ exact attention they replace; since the ratio between clustering and monopole cost is fixed in $N$ and $C$, clustering cannot overtake the rest of the approximation as either grows. Consistent with this, the clustering share of total runtime remains at 7–8% as the cluster count doubles from 64 to 128 to 256 at constant retrieval fraction.

**Behaviour Early in Training**    MuSe is applied from the first training step, with no warmup. Early in training the learned representations are still forming, yet K-means recovers substantial cluster structure from them, enough for the resulting partitions to be useful. The higher-entropy attention patterns characteristic of early training are moreover favourable for the monopole component, since smoother attention is captured more accurately by a cluster summary. Together these let MuSe track exact attention from the start (Figure 6).

# D. Hyperparameters

**Model Architecture**   We use a standard decoder-only transformer with pre-norm (layer normalization before attention and FFN), RoPE position encodings with base frequency $10^5$, and embedding tying. Model sizes are parameterized by a scale factor $S \in \{3, 4, 5, 6, 8\}$ corresponding to $\{96M, 185M, 350M, 560M, 1B\}$ parameters:

- Embedding dimension: $256 \times S$

- Attention heads: $4 \times S$ (head dimension 64)

- FFN dimension: $1024 \times S$

- Layers: 10, 12, 16, 18, 20 for scales 3–8 ($\approx 3S$, rounded to even for layer pairing; scale 8 uses 20 rather than 24 to hit the target parameter count)

- Vocabulary size: 32768

- Context length: 64k tokens

Layers are grouped into super blocks of 2: one local attention layer (sliding window of 256 tokens) followed by one global attention layer. FFN uses ReLU activation. Layer normalization uses $\epsilon = 10^{-6}$.

**Optimizer**   We use AdamW with $\beta_1 = 0.9$, $\beta_2 = 0.95$, $\epsilon = 10^{-6}$, and weight decay 0.01. Gradients are clipped to global norm 1.0. Learning rate follows a warmup-cosine schedule with 4000 warmup steps and final value 10% of peak. Peak learning rate is $3 \times 10^{-3}$, scaled by $S^{-0.5}$.

**Training Data**   We train on two datasets:

- **Code:** The Stack v2 (Lozhkov et al., 2024) (`bigcode/the-stack-v2-train-smol-ids`). We filter to Python files, excluding vendor and generated code. Repositories are bucketed by estimated token count (Python bytes / 3.5) on a $\log_2$ scale; we use buckets 5–8 corresponding to 16k–256k tokens per repository. Files within each repository are concatenated with `<reponame>` and `<filename>` delimiters, sorted by path depth. Total: 21.5B tokens.

- **Scientific documents:** OLMo 3 Long-Mino pool (Team Olmo et al., 2025) (`allenai/dolma3_longmino_pool`), `science_tech` topic, bucket 2e16 (64k–128k tokens per document). Total: 37.5B tokens.

Both datasets use separate BPE tokenizers with 32768 vocabulary trained on domain-specific samples. Training token counts scale with model size: 2.0B, 4.0B, 7.5B, 12.9B, and 23.6B tokens for scales 3–8 respectively. Since the code

corpus totals 21.5B tokens, the largest code run (scale 8) slightly exceeds one epoch ($\sim 1.1\times$); all other runs remain within a single epoch of their corpus. Batch size is 2 sequences of 64k tokens (128k tokens per step).

**MuSe Operating Point**   For pretraining experiments, we use 128 query clusters, 128 key clusters, top-8 cluster retrieval, top-1 spatial block retrieval, and 8k spatial blocks (Q128K128R8SP1B8k). This configuration achieves <1% relative squared error on microbenchmarks while providing $2\times$ speedup over CUDNN Flash Attention.

**Hardware**   All pretraining experiments use $8\times$ NVIDIA A100 (80GB) GPUs in a $(2, 4)$ mesh with data parallelism over the batch dimension and model parallelism over attention heads. Training uses mixed precision (bfloat16/float32).

**Llama Evaluation**   For generalization experiments (Section 4.3), we evaluate on pretrained Llama 3.2 1B and Llama 3.1 8B models converted to Flax. We measure per-position cross-entropy loss on PG-19 (Rae et al., 2020) (test split) with 64k context, using 100 document chunks. Cumulative mean loss is computed across positions and plotted as reducible loss (subtracting fitted irreducible loss).

**Approximation Quality Metric**   We report relative squared error (RSE) as our primary measure of approximation quality: $\text{RSE} = \|o_{\text{approx}} - o_{\text{exact}}\|^2 / \|o_{\text{exact}}\|^2$, where $o$ denotes the attention output, averaged across all queries in the evaluation batch.

# E. 96M Model Microbenchmarks

This appendix presents detailed microbenchmark results from a 96M parameter model. We evaluate approximation quality on queries, keys, and values extracted from the trained model with 64k context. We use 2 sequences across all 60 attention heads (from all layers), with head dimension 64. Our base configuration uses 128 query clusters, 128 key clusters, top-8 cluster retrieval, top-1 spatial block retrieval, and 8k spatial blocks (Q128K128R8SP1B8k).

## E.1. Speedup vs Exact Attention

Table 12 compares MuSe to exact flash attention implementations. At our base configuration (QK128R8), MuSe achieves $2\times$ speedup over CUDNN Flash Attention with 2.6% relative squared error.

## E.2. Effect of Query Clustering

Table 13 shows the effect of varying the number of query clusters while holding other parameters fixed. Reducing from 128 to 1 query cluster (mathematically equivalent to no query clustering) increases relative squared error by

*Table 12.* Runtime comparison with exact attention (64k context, head dim 64, 96M model).

| Method | Time (ms) | Speedup | RSE |
|---|---|---|---|
| CUDNN Flash | 212.1 | 1.0× | 0 (exact) |
| Pallas Flash | 243.6 | 0.87× | 0 (exact) |
| MuSe QK64R8 | 118.1 | 1.80× | 0.0315 |
| MuSe QK128R8 | 105.5 | 2.01× | 0.0259 |
| MuSe QK256R8 | 123.3 | 1.72× | 0.0206 |

1.9× (0.026 to 0.049) with negligible runtime difference, validating the importance of query clustering.

*Table 13.* Effect of query cluster count (K128R8SP1B8k fixed). Q1 uses zeroed centroids, mathematically equivalent to no query clustering.

| Q | RSE | Correlation | Time (ms) |
|---|---|---|---|
| 256 | 0.0214 | 0.9875 | 122.6 |
| 128 | 0.0259 | 0.9848 | 105.5 |
| 64 | 0.0301 | 0.9825 | 104.3 |
| 32 | 0.0338 | 0.9764 | 102.0 |
| 16 | 0.0376 | 0.9737 | 103.6 |
| 1* | 0.0494 | 0.9685 | 105.6 |

### E.3. Effect of Key Clustering

Table 14 shows the effect of varying key clusters with retrieval fraction held constant (R/K = 1/16). More clusters with proportionally more retrieval improves accuracy, with K128R8 providing a good speed/accuracy tradeoff.

*Table 14.* Effect of key cluster count with fixed retrieval fraction (Q128SP1B8k fixed, R/K = 1/16).

| K | R | RSE | Correlation | Time (ms) |
|---|---|---|---|---|
| 256 | 16 | 0.0180 | 0.9882 | 137.1 |
| 128 | 8 | 0.0259 | 0.9848 | 105.5 |
| 64 | 4 | 0.0359 | 0.9762 | 106.7 |
| 32 | 2 | 0.0504 | 0.9639 | 117.8 |
| 16 | 1 | 0.0692 | 0.9539 | 126.1 |

### E.4. Joint Query and Key Clustering

Table 15 compares performance with and without query clustering as the number of key clusters varies. Query clustering provides 1.4–2.4× error reduction, with larger benefits at higher cluster counts where query-specific summaries matter more.

### E.5. Monopole Only (No Retrieval)

Table 16 shows performance without retrieval, comparing monopole approximation with and without query clustering. Query clustering provides 1.9–2.7× error reduction for monopole-only—larger than the 1.4–2.4× benefit observed

*Table 15.* Effect of query clustering across key cluster counts (R/K = 1/16, SP1B8k fixed). "No Q" uses zeroed query centroids.

| K | R | RSE | | Corr | | Gain |
|---|---|---|---|---|---|---|
| | | No Q | With Q | No Q | With Q | |
| 256 | 16 | 0.0365 | 0.0152 | 0.9749 | 0.9891 | 2.4× |
| 128 | 8 | 0.0494 | 0.0259 | 0.9685 | 0.9848 | 1.9× |
| 64 | 4 | 0.0676 | 0.0414 | 0.9541 | 0.9723 | 1.6× |
| 32 | 2 | 0.0943 | 0.0640 | 0.9400 | 0.9576 | 1.5× |
| 16 | 1 | 0.1295 | 0.0933 | 0.9177 | 0.9373 | 1.4× |

with retrieval (Table 15). This confirms that query clustering is most critical for the monopole component; retrieval partially masks the benefit by handling high-attention clusters exactly.

*Table 16.* Monopole only (no retrieval), with and without query clustering. "No Q" uses zeroed query centroids.

| QK | RSE | | Corr | | Gain |
|---|---|---|---|---|---|
| | No Q | With Q | No Q | With Q | |
| 256 | 0.1560 | 0.0586 | 0.9005 | 0.9606 | 2.7× |
| 128 | 0.1812 | 0.0794 | 0.8879 | 0.9430 | 2.3× |
| 64 | 0.2118 | 0.1025 | 0.8697 | 0.9283 | 2.1× |
| 32 | 0.2362 | 0.1273 | 0.8578 | 0.9131 | 1.9× |

### E.6. Cluster Count vs Retrieval Work

Table 17 shows a striking result: increasing cluster count improves accuracy while *reducing* retrieval work. With R=8 fixed, each doubling of cluster count improves error by 1.2–1.3× while halving the number of keys retrieved per query. QK512R8 achieves 2× better accuracy than QK64R8 while retrieving 8× fewer keys. This demonstrates that finer clustering improves monopole quality enough to more than compensate for retrieving from smaller clusters.

Table 18 shows the effect of varying retrieval count with cluster count fixed. Doubling retrieval from R8 to R16 improves error by 1.27×—comparable to the 1.26× improvement from doubling cluster count (QK128 to QK256) at fixed R=8.

*Table 17.* Effect of cluster count with fixed retrieval count R=8 (SP1B8k fixed). Higher cluster counts improve accuracy despite less retrieval work.

| QK | R | RSE | Correlation | Keys/query | Time (ms) |
|---|---|---|---|---|---|
| 512 | 8 | 0.0156 | 0.9908 | 1024 | 226.0 |
| 256 | 8 | 0.0206 | 0.9887 | 2048 | 123.3 |
| 128 | 8 | 0.0259 | 0.9848 | 4096 | 105.5 |
| 64 | 8 | 0.0315 | 0.9789 | 8192 | 118.1 |

*Table 18.* Effect of retrieval count with fixed cluster count (QK128SP1B8k fixed).

| QK | R | RSE | Correlation | Time (ms) |
|-----|-----|--------|-------------|-----------|
| 128 | 16 | 0.0204 | 0.9851 | 140.8 |
| 128 | 8 | 0.0259 | 0.9848 | 105.6 |
| 128 | 4 | 0.0355 | 0.9739 | 89.6 |

### E.7. Retrieval Only (No Monopole)

Table 19 shows performance with retrieval only, disabling the monopole contribution. Query clustering provides only $1.07\times$ benefit for retrieval selection, compared to $2.3\times$ for monopole-only (Table 16). This confirms that query clustering primarily improves monopole quality; the retrieval mechanism selects reasonable clusters even without query-specific summaries. Importantly, query clustering does not harm retrieval, making it a pure win.

*Table 19.* Retrieval only (no monopole) at QK128R8SP1B8k. "No Q" uses zeroed query centroids.

| Setting | RSE | Correlation | Time (ms) |
|-----------|--------|-------------|-----------|
| With Q | 0.0990 | 0.9645 | 103.4 |
| No Q | 0.1061 | 0.9556 | 103.2 |
| Q benefit | | $1.07\times$ | |

## F. 1B Model Microbenchmarks

This appendix presents detailed microbenchmark results from our 1B parameter model (320 heads, head dimension 64, 64k context). These experiments validate hyperparameter choices and characterize approximation quality at scale.

### F.1. Block Size and Spatial Retrieval Tradeoff

Table 20 shows the effect of varying block size $B$ while adjusting spatial retrieval $SP$ to maintain constant retrieval span ($SP \times B = 16k$). With cluster count and cluster retrieval fixed (QK128R4), this sweep illustrates the fundamental tradeoff between exact local attention and approximate far-field attention.

*Table 20.* Effect of block size with constant retrieval span (QK128R4, $SP \times B = 16k$, 1B model, 40 heads, 2 sequences). Larger blocks compute more attention exactly but have higher diagonal cost; smaller blocks approximate more attention but eventually become slow due to accumulation overhead.

| Config | RSE | Correlation | Time (ms) |
|------------------|---------|-------------|-----------|
| QK128R4SP1B16k | 0.00493 | 0.9974 | 105.6 |
| QK128R4SP2B8k | 0.00704 | 0.9959 | 75.5 |
| QK128R4SP4B4k | 0.01578 | 0.9901 | 71.8 |
| QK128R4SP8B2k | 0.03833 | 0.9767 | 129.4 |

Accuracy degrades as block size decreases ($0.00493 \rightarrow$

0.03833, a $7.8\times$ increase in error) because more of the attention mass—which is predominantly local—must be routed through the approximation rather than computed exactly in the block diagonal.

Runtime exhibits a U-shape: $B = 4k$–8k is fastest ($\sim$72–76ms), while both $B = 16k$ (expensive diagonal attention) and $B = 2k$ (expensive accumulation over 32 blocks plus 8 spatial selections per query) are slower. This demonstrates that block size provides a tunable speed/accuracy tradeoff, with $B = 8k$ offering a good balance for 64k context.

We use $SP = 1$ (single spatial block per cluster) for simplicity; $SP = 2$ with reduced cluster retrieval ($R = 4$) shows similar microbenchmark performance but exhibited more adaptation during pretraining.

### F.2. Semantic vs. Spatial Retrieval Breadth

The far-field retrieval budget is the number of (cluster, spatial-block) pairs attended exactly per query, $k_1 \cdot k_2$, where $k_1$ (denoted $R$) is the number of retrieved clusters and $k_2$ (denoted $SP$) the number of spatial blocks retrieved within each cluster. A single retrieval count $k = k_1 \cdot k_2$ over the joint space would suffice if quality depended only on the product; separating $k_1$ and $k_2$ is justified only if the split itself matters. Table 21 varies the split at a fixed budget of eight pairs (QK128, $B$=8k): shifting budget from semantic breadth ($k_1$) toward spatial breadth ($k_2$) degrades approximation quality, with relative squared error nearly tripling from the semantic-heavy R8SP1 (our operating point) to the spatial-heavy R2SP4. The R4SP2 configuration coincides with the $SP2B8k$ row of Table 20.

*Table 21.* Semantic vs. spatial retrieval breadth at a fixed far-field budget of $k_1 \cdot k_2 = 8$ pairs per query (1B model, QK128, $B$=8k, 64k context, 40 heads). Semantic breadth ($k_1$) dominates spatial breadth ($k_2$) on our data.

| Config | $k_1$ ($R$) | $k_2$ ($SP$) | RSE | Correlation |
|--------|---------|----------|-------------|-------------|
| R8SP1 | 8 | 1 | **0.00622** | 0.9965 |
| R4SP2 | 4 | 2 | 0.00704 | 0.9959 |
| R2SP4 | 2 | 4 | 0.01797 | 0.9892 |

The configuration in which spatial breadth would dominate is narrow. Retrieval is independent per token and per head, so different tokens or heads needing spatially distant information is already handled. The remaining failure mode requires a single token, on a single head, to need fine-grained access to specific keys at positions separated by whole documents that fall in different spatial blocks of the same semantic cluster, and even then only when simple lookup does not suffice (any copy of the relevant key being adequate), i.e., when fine-grained non-redundant aggregation across those blocks is required. We found semantic breadth to dominate on our data; on a domain where such

dispersed aggregations proved important, shifting budget from $k_1$ to $k_2$ would be the natural adjustment, which our implementation supports.

## F.3. Cluster Count with Fixed Retrieval

Table 22 shows the effect of varying cluster count with retrieval count held fixed at R=8. Higher cluster counts improve accuracy ($0.00838 \rightarrow 0.00456$, a $1.8\times$ error reduction) while reducing retrieval work ($8192 \rightarrow 2048$ keys per query). This confirms the 96M result (Table 17): finer clustering improves monopole quality enough to more than compensate for retrieving from smaller clusters.

*Table 22.* Effect of cluster count with fixed retrieval R=8 (SP1B8k, 1B model, 80 heads, 2 sequences, 64k context).

| Config | RSE | Correlation | Keys/query |
|---|---|---|---|
| QK64R8 | 0.00838 | 0.9949 | 8192 |
| QK128R8 | 0.00622 | 0.9965 | 4096 |
| QK256R8 | 0.00456 | 0.9975 | 2048 |

## F.4. Monopole Runtime Scaling

We measure monopole-only runtime (no retrieval) to characterize the scaling of the novel MuSe components: clustering, summary computation, and causal accumulation. All timings are for 40 heads on a single node (5 heads per A100).

**Sequence Length at Constant Total Tokens**   Table 23 shows monopole runtime at constant total tokens (128k) but varying sequence length and batch size. Longer sequences are modestly slower despite equal total work, likely due to reduced parallelism across sequences or less efficient clustering on larger point sets rather than algorithmic overhead.

*Table 23.* Monopole-only runtime at constant total tokens (128k), varying sequence length (QK128B8k, 40 heads).

| Config | Sequences | Blocks/seq | Time (ms) |
|---|---|---|---|
| N=32k | 4 | 4 | 33.3 |
| N=64k | 2 | 8 | 36.0 |
| N=128k | 1 | 16 | 41.0 |

Per-token cost increases modestly with sequence length: 0.26 $\mu$s/token at 32k context to 0.32 $\mu$s/token at 128k context (+23%). This is slightly super-linear in sequence length at constant batch size, but the overhead is mild.

**Cluster Count Scaling**   Table 24 shows monopole runtime as cluster count varies at fixed context length. Increasing clusters from 64 to 256 ($4\times$) increases runtime by only 31%, indicating that cluster-dependent costs (summaries, accumulation) are not dominant.

If accumulation (which scales as $C^2$) dominated, we would

*Table 24.* Monopole-only runtime varying cluster count (B=16k, N=64k, 40 heads, constant total tokens via batching).

| Clusters | Time (ms) | vs QK64 |
|---|---|---|
| QK64 | 52.1 | — |
| QK128 | 55.3 | +6% |
| QK256 | 68.3 | +31% |

expect $4\times$ clusters to yield $16\times$ runtime; the observed 31% increase confirms that accumulation is a small fraction of total monopole cost.

**Sequence Length with Larger Blocks**   Table 25 shows a similar experiment with B=16k blocks and QK64.

*Table 25.* Monopole-only runtime at constant total tokens (128k), varying sequence length (QK64B16k, 40 heads).

| Config | Sequences | Blocks/seq | Time (ms) |
|---|---|---|---|
| N=32k | 4 | 2 | 49.4 |
| N=64k | 2 | 4 | 52.1 |
| N=128k | 1 | 8 | 57.9 |

The pattern is consistent: longer sequences are slower at constant total tokens (+17% from 32k to 128k), but the overhead is modest and likely attributable to reduced parallelism or clustering efficiency rather than algorithmic complexity.

## F.5. Comparison with MoBA

The main text compares MuSe against MoBA at matched far-field sparsity (8k block diagonal, $64\times$ sparsity), isolating the quality of far-field approximation. For completeness, we also compare against MoBA with settings closer to typical usage: 512-token block diagonal with top-8 retrieval from 512-token blocks ($8\times$ far-field sparsity). This configuration allocates more compute budget to far-field retrieval at the cost of local attention quality.

Table 26 shows that even under these favorable conditions, MuSe maintains an advantage at both scales. Note that this comparison favors MoBA: our 8k block size is chosen to balance accuracy and runtime with our kernels, not to maximize sparsity. The 512-block MoBA configuration would be substantially slower than exact attention when implemented with our kernels.

## F.6. Far-Field Scaling with Context Length

Table 27 shows approximation quality versus context length with scaled hyperparameters: QK $\propto \sqrt{N}$, R $\propto \sqrt{N}$ (constant retrieval fraction), $B \propto N$ (constant 8 spatial blocks). This maintains $64\times$ far-field sparsity and 1/8 block diagonal fraction across all context lengths.

Approximation error decreases substantially with scale

*Table 26.* Extended MoBA comparison at 185M and 560M scale. "Sparsity-matched" uses 8k block diagonal with 64× far-field sparsity; "Budget-matched" uses 512-token blocks with 8× far-field sparsity. Cross-entropy loss, lower is better.

| Scale | Method | Test Attention | |
|---|---|---|---|
| | | CUDNN | Method |
| 185M | CUDNN (baseline) | 0.9400 | — |
| | MuSe | **0.9384** | 0.9435 |
| | MoBA (sparsity-matched) | 0.9714 | 1.0027 |
| | MoBA (budget-matched) | 0.9460 | 0.9633 |
| 560M | CUDNN (baseline) | 0.7745 | — |
| | MuSe | **0.7728** | 0.7746 |
| | MoBA (sparsity-matched) | 0.7958 | 0.8209 |
| | MoBA (budget-matched) | 0.7815 | 0.7858 |

*Table 27.* Far-field approximation quality vs. context length with scaled hyperparameters (8 spatial blocks throughout). Relative squared error decreases as context grows. *128k context created by concatenating two 64k sequences.

| $N$ | QK | $R$ | $B$ | RSE | Cosine |
|---|---|---|---|---|---|
| *Series 1:* | | | | | |
| 4k | 32 | 2 | 512 | 0.00884 | 0.9947 |
| 16k | 64 | 4 | 2k | 0.00662 | 0.9958 |
| 64k | 128 | 8 | 8k | 0.00622 | 0.9965 |
| *Series 2:* | | | | | |
| 2k | 16 | 1 | 256 | 0.01701 | 0.9899 |
| 8k | 32 | 2 | 1k | 0.01096 | 0.9929 |
| 32k | 64 | 4 | 4k | 0.00863 | 0.9946 |
| 128k* | 128 | 8 | 16k | 0.00723 | 0.9956 |

$(0.017 \rightarrow 0.007$ from 2k to 128k), implying that the compute required to achieve a given error level grows subquadratically in the far-field. We do not determine the precise exponent, which would likely be data-dependent. The block diagonal cost remains $O(N^2/S)$; for very long contexts where this dominates, recursive application of MuSe would reduce it further.

# G. Additional Results

## G.1. Effective Cluster Count Comparison

Tables 28 and 29 compare MuSe and the no-query-clustering ablation by computing effective cluster counts. For each configuration, we interpolate (or extrapolate) using a power-law fit to determine what cluster count the other method would require to achieve the same error. Both methods follow approximate power laws: MuSe with exponent $d \approx -0.89$ and no-query-clustering with $d \approx -0.59$. The steeper slope for MuSe means its advantage grows with cluster count.

*Table 28.* MuSe configurations with equivalent no-query-clustering cluster counts. R = QK/16 throughout. Effective cluster count interpolated via power-law fit. Multiplier shows how many more clusters the no-query-clustering method would need.

| QK | RSE | Corr | Eff. No-Q | Mult. |
|---|---|---|---|---|
| 16 | 0.0380 | 0.9775 | 82 | 5.1× |
| 32 | 0.0210 | 0.9878 | 191 | 6.0× |
| 64 | 0.0114 | 0.9935 | 568 | 8.9× |
| 128 | 0.0062 | 0.9965 | 1179 | 9.2× |
| 256 | 0.0033 | 0.9980 | 2518 | 9.8× |
| 512 | 0.0017 | 0.9990 | 5481 | 10.7× |

*Table 29.* No-query-clustering configurations with equivalent MuSe cluster counts. R = QK/16 throughout. Multiplier shows how many fewer clusters MuSe needs to achieve the same error.

| QK | RSE | Corr | Eff. MuSe | Mult. |
|---|---|---|---|---|
| 16 | 0.0991 | 0.9464 | 6 | 2.7× |
| 32 | 0.0665 | 0.9640 | 9 | 3.6× |
| 64 | 0.0428 | 0.9775 | 13 | 4.9× |
| 128 | 0.0284 | 0.9856 | 19 | 6.7× |
| 256 | 0.0191 | 0.9904 | 36 | 7.1× |
| 512 | 0.0129 | 0.9935 | 57 | 9.0× |

## G.2. Downstream Evaluation

To verify that MuSe does not degrade downstream task performance, we evaluate 1B models trained on scientific PDFs using the lm-eval harness on ARC-Easy and SciQ. Table 30 shows that MuSe-trained models perform comparably to CUDNN-trained models, with all differences within one standard error. Both models are well above the 25% random baseline, confirming that the approximation does not impair learned representations.

*Table 30.* Downstream evaluation on ARC-Easy and SciQ (1B model trained on scientific PDFs). All differences are within standard error ($\sim$0.01).

| Benchmark | Metric | CUDNN | MuSe |
|---|---|---|---|
| ARC-Easy | acc | 0.479 | 0.487 |
| ARC-Easy | acc_norm | 0.431 | 0.436 |
| SciQ | acc | 0.766 | 0.759 |
| SciQ | acc_norm | 0.663 | 0.659 |

## G.3. RULER Results

Table 31 reports per-task RULER accuracy at 64k context, for both domains and both training methods. As discussed in Section 4.3, both MuSe- and CUDNN-trained models score poorly on most variants (the code models near zero on all but single-needle retrieval), which we attribute to domain shift and the limited instruction-following of our small base models rather than to a retrieval deficit.

*Table 31.* **Per-task RULER NIAH accuracy at 64k context.** Accuracy $\pm$ binomial std. error, $n = 500$ per cell. lm-eval-harness reports no stderr for these tasks; we compute $\sqrt{p(1-p)/n}$. Both methods were run on all tasks for both domains. See Section 4.3 for interpretation.

| Task | CUDNN | MuSe |
|---|---|---|
| *Scientific PDF domain* | | |
| niah_single_1 | $61.0 \pm 2.2\%$ | $\mathbf{85.0 \pm 1.6\%}$ |
| niah_single_2 | $\mathbf{46.6 \pm 2.2\%}$ | $26.6 \pm 2.0\%$ |
| niah_multikey_1 | $\mathbf{50.4 \pm 2.2\%}$ | $28.0 \pm 2.0\%$ |
| niah_multiquery | $\mathbf{23.8 \pm 1.9\%}$ | $15.4 \pm 1.6\%$ |
| niah_multivalue | $\mathbf{20.9 \pm 1.8\%}$ | $15.9 \pm 1.6\%$ |
| *Code domain* | | |
| niah_single_1 | $62.8 \pm 2.2\%$ | $\mathbf{70.6 \pm 2.0\%}$ |
| niah_single_2 | $0.0\%$ | $1.4 \pm 0.5\%$ |
| niah_multikey_1 | $0.0\%$ | $2.2 \pm 0.7\%$ |
| niah_multiquery | $0.0\%$ | $0.0\%$ |
| niah_multivalue | $4.6 \pm 0.9\%$ | $0.0\%$ |

### G.4. Custom NIAH Accuracy by Needle Depth

Table 32 resolves the custom-NIAH uid-level exact match of Section 4.3 by needle depth, for the fixed-depth variants (single-needle and multi-key; the multi-query variants place needles at randomly sampled depths and so are not bucketed here). MuSe-trained accuracy remains above 89% at every depth. The exact-attention baseline, by contrast, degrades sharply toward the middle depths on the harder multi-key $4\times$ variant (63.6% at depth 0.25 versus 96.7% at 0.75).

*Table 32.* Custom code NIAH uid-level exact match (%) by needle-depth bucket, for the fixed-depth variants. CUDNN denotes exact-attention training and MuSe denotes MuSe training; all evaluation uses exact attention.

| | Single-needle | | Multi-key $2\times$ | | Multi-key $4\times$ | |
|---|---|---|---|---|---|---|
| Depth | CUDNN | MuSe | CUDNN | MuSe | CUDNN | MuSe |
| 0.05 | 96.8 | 89.1 | 92.0 | 89.8 | 85.1 | 90.7 |
| 0.15 | 97.5 | 98.6 | 91.6 | 99.6 | 77.3 | 98.1 |
| 0.25 | 98.2 | 99.7 | 86.2 | 100.0 | 63.6 | 98.9 |
| 0.35 | 97.2 | 100.0 | 84.4 | 100.0 | 67.3 | 99.3 |
| 0.45 | 97.9 | 100.0 | 90.2 | 99.6 | 75.5 | 98.9 |
| 0.55 | 97.5 | 99.3 | 93.1 | 98.9 | 83.3 | 97.0 |
| 0.65 | 97.5 | 99.3 | 98.6 | 98.2 | 90.7 | 94.8 |
| 0.75 | 99.3 | 99.7 | 99.3 | 100.0 | 96.7 | 98.5 |
| 0.85 | 98.2 | 99.7 | 99.3 | 100.0 | 94.1 | 100.0 |

### G.5. Fine-tuning to Remove Adaptation

At 1B scale on the code domain, the MuSe-trained model shows minor adaptation to the approximation: when evaluated with CUDNN attention, loss is 0.7108 compared to the baseline of 0.7026. We investigate whether brief fine-tuning with exact attention removes this adaptation.

Table 33 shows the progression of MuSe→CUDNN evaluation loss during CUDNN fine-tuning. The adaptation gap

closes rapidly: after just 26M tokens (0.1% of pretraining), the model not only matches but *beats* the CUDNN baseline (0.6994 vs 0.7026). Continued fine-tuning yields diminishing returns, with 93% of the improvement occurring in the first 0.1% of fine-tuning tokens. This confirms that adaptation effects are shallow and easily removed, making the recommended workflow straightforward: pretrain with MuSe for speedup, then briefly fine-tune with exact attention before deployment. As a preliminary alternative that avoids the fine-tuning step entirely, the "Aggregation Weight Gradient Flow" paragraph of Appendix C reports a training-time gradient modification that removes the adaptation gap directly.

*Table 33.* Fine-tuning progression on 1B code model. MuSe→CUDNN evaluation loss during CUDNN fine-tuning.

| MTok | % Pretrain | Loss | Note |
|---|---|---|---|
| 0 | 0% | 0.7108 | Before fine-tuning |
| 26 | 0.1% | 0.6994 | Beats baseline (0.7026) |
| 210 | 0.9% | 0.6985 | Continued improvement |

## H. Distributed Training Considerations

MuSe is compatible with standard distributed training strategies.

**Sequence parallelism** In Megatron-style sequence parallelism, attention heads are sharded across devices, with the full sequence gathered for each head before attention. From the perspective of MuSe, each device simply approximates fewer heads over the complete sequence—the distributed origin of sequence fragments is invisible to the approximation. Our pretraining experiments use this approach.

**Ring attention** Ring attention distributes the sequence across devices, rotating key/value blocks around a ring. The block-diagonal (local, causal) attention is naturally handled by MuSe's spatial blocking. For off-diagonal (remote, non-causal) blocks, MuSe extends naturally: either retain spatial blocking with the causal mask removed, or treat remote context as a single spatial block with purely semantic sparsity. We leave detailed investigation of these extensions to future work.

## I. Code Availability

The research code used for the experiments in this paper is available as the supplementary material accompanying the submission on OpenReview. It is provided to document the exact implementation used for the experiments reported here, and for reproducibility, not as a maintained or general-purpose implementation of the method.

