# OpenReview forum: "Multipole Semantic Attention: A Fast Approximation of Softmax Attention for Pretraining"
_ICML.cc/2026/Conference — ICML 2026 regular_

### Official Review · Reviewer_KhBo · 2026-02-17

**Soundness:** 3
**Presentation:** 4
**Significance:** 3
**Originality:** 4
**Overall Recommendation:** 5
**Confidence:** 2

**Summary:**

This paper is about Mulitpole Semantic Attention (MuSe) - a fast approximation of attention - meant to reduce the O(N^2) bottleneck of attention. MuSe clusters both keys and queries separately in the representation space, allowing them to let coarse query clusters attend to fine key clusters to create query-specific exponentially tilted summaries, followed by a retrieval step where queries attend to the most relevant key clusters.

The authors evaluate MuSe on a 1B parameter model with a 64k context window and see a 36% speed up during pretraining without much, if any, loss in performance.

**Compliance With Llm Reviewing Policy:**

Affirmed.

**Key Questions For Authors:**

1. How sensitive is MuSe to the choice of cluster size? Would I see stark performance drops if I wasn't careful about this choice?

2. From my understanding, you used MuSe after a warmup period. How would MuSe perform early in training before representations become semantically meaningful?

3. In the limitations, you mention reusing key/value summaries across query heads for Grouped-Query Attention (GQA). Given that Llama models rely heavily on GQA, were the summaries computed redundantly per query head in your experiments?

**Limitations:**

yes

**Strengths And Weaknesses:**

Soundness:

The paper is technically sound and supported by extensive empirical validation at scale. They show that pretraining using MuSe on both code repos and technical documents speeds up the process without a performance drop. They have clear metrics showing their method's dominance over reasonable benchmarks. The few weaknesses are clearly denoted. The big one is that they test this on relatively small models, especially with the head dimension being 64 instead of 128.

Presentation:

The paper is clearly written, well structured, and well-positioned amongst prior work

Significance:

They tackle a meaningful and relevant problem. Attention is the computational bottleneck in LLM training and fast approximations have already proven wildly successful with work like Flash Attention. The only concern I have is that this work is unlikely to see adoption without more technical support because of the engineering difficulty of implementing MuSe combined with the lack of kernel optimization; however , that is a relatively small issue.

Originality:

This work adds a novel dimension to attention approximations with a creative physics-inspired influence.

---

> ### Author Rebuttal · Authors · 2026-03-30
>
> We thank Reviewer KhBo for their positive and thoughtful assessment.
>
> **Cluster size sensitivity.** MuSe exhibits smooth, predictable degradation as cluster count varies. Table 2 in the paper shows relative squared error following a power law from QK16 (0.038) to QK512 (0.002), with no cliffs or sharp transitions. Our operating point (QK128) was chosen as a speed/quality tradeoff, not because nearby settings fail — the same hyperparameters are used across our entire scaling analysis (96M to 1B) without modification, confirming robustness.
>
> **MuSe early in training.** We use MuSe from step 0 with no warmup period. Figure 6 in the paper shows that MuSe closely tracks CUDNN throughout training from 0.5B to 21.5B tokens, confirming the approximation works from the start. Early representations already have enough structure for K-means to produce useful partitions; moreover, early attention patterns have higher entropy and lower query magnitudes, which is the regime where the monopole approximation is tightest.
>
> **GQA.** Yes, MuSe applies to GQA with no modification and identical relative speedup: MuSe simply runs independently for every query head and associated kv head, exactly mirroring how exact attention handles GQA. The paper mentions GQA in the context of future work (not limitations): reusing query-dependent summaries across query heads within the same group would be an additional optimization beyond what we already demonstrate, not a requirement for the method to work.
>
> **Scale and head dimension.** For context, our 64k training is roughly 8x more expensive in attention flops per token than the 8k used by most comparable methods, placing our compute investment comparable with the most expensive from-scratch pretraining runs in the related works. While we share with most comparable works the limitation of focusing on one head dimension for our scaling sweep, we note that our choice of head dim 64 follows frontier practice (e.g., OpenAI's 20B and 120B GPT-OSS models), and we additionally validate head dim 128 via pretraining at 185M (matching the baseline) and Llama 3.1-8B generalization experiments.
>
> **Additional evaluations.** Following suggestions from our reviewers, we have conducted long-context retrieval evaluations (RULER NIAH, custom code NIAH with multi-key and multi-query variants, and LongBench code completion) at 64k context. MuSe-trained models match or exceed exact-attention baselines on in-distribution downstream and retrieval tasks, including multi-key and multi-query variants (e.g. 97.4% vs 81.5% for K=4), see response to Reviewer AkdV for details.

---

> > ### Author Rebuttal · Reviewer_KhBo · 2026-03-31
> >
> > They've answered my questions. I see no reason to change my score - 5. However, I would reiterate to the AC that I lack confidence in my assessment, as this is not my field.

---

### Official Review · Reviewer_5Eph · 2026-03-09

**Soundness:** 2
**Presentation:** 3
**Significance:** 2
**Originality:** 2
**Overall Recommendation:** 4
**Confidence:** 4

**Summary:**

This paper introduces Multipole Semantic Attention (MuSe), a novel approximation mechanism for softmax attention designed to efficiently train and deploy large language models with extremely long contexts (e.g., 64k tokens). Inspired by the Fast Multipole Method, the core innovation is to cluster both queries and keys separately in their semantic space, rather than relying solely on spatial position. This allows for the computation of query-specific cluster summaries (monopole approximation), significantly improving accuracy. To correct for residual errors, MuSe employs a two-level retrieval strategy that selects important clusters and, within them, important spatial blocks for exact token-level attention. The method achieves substantial efficiency gains, demonstrating a 2× speedup over cuDNN Flash Attention and a 36% improvement in end-to-end pre-training throughput for a 1B model at 64k context. Crucially, MuSe is compatible with pre-trained models like Llama due to its use of exponential tilting, and models trained with it can switch to exact attention at test time with minimal adaptation. Extensive experiments validate that MuSe maintains competitive loss and downstream task performance while providing significant computational savings.

**Compliance With Llm Reviewing Policy:**

Affirmed.

**Key Questions For Authors:**

1. Regarding the Overhead and Scalability of the Clustering Algorithm:The paper mentions using streaming K-means for clustering and states that its overhead is acceptable. Could you please clarify the approximate proportion of total computation (FLOPs) or runtime occupied by the clustering algorithm itself during training? As the context length N or the number of clusters C increases further, could the clustering step become a new bottleneck? Could the authors provide a computational complexity analysis for the clustering algorithm?
2. Regarding the Robustness of the Retrieval Mechanism:MuSe relies on a two-level retrieval mechanism k1 clusters, k2 blocks to capture residual interactions. How robust is this retrieval mechanism to "attention patterns" unseen during training? For instance, in scenarios requiring attention to numerous but widely dispersed key pieces of information within a long context (i.e., where attention is neither extremely sharp nor perfectly smooth), might the fixed k1 and k2 parameters miss important information?

**Limitations:**

Yes

**Strengths And Weaknesses:**

Soundness: The paper is technically sound, with well-designed experiments and thorough ablation studies that validate the core claims. The authors carefully evaluate the contribution of each component (query clustering, exponential tilting, two-level retrieval) and provide honest comparisons with baselines like MoBA.
Presentation:The paper is clearly written and well-structured, with intuitive geometric explanations and a logical flow from motivation to implementation to experiments. The authors effectively position their work in the context of prior literature, particularly drawing connections to the Fast Multipole Method and sparse attention mechanisms. However, the main text occasionally defers important details to the appendix (e.g., clustering algorithms, dipole derivation), which may disrupt readability for readers seeking a complete understanding without consulting supplementary material.
Significance:This work addresses the important and timely problem of efficient long-context processing in LLMs. The 2× speedup and 36% throughput improvement at 64k context represent meaningful practical gains, and the compatibility with pretrained models (Llama) enhances real-world applicability. The approach could influence future research on approximate attention mechanisms, particularly the insight that query clustering provides substantial benefits at negligible cost. However, the significance is somewhat tempered by the focus on head dimension 64 and the early-stage nature of some results (e.g., limited downstream task evaluation).
Originality: The paper demonstrates strong originality through its creative combination of ideas: adapting the Fast Multipole Method to semantic space, introducing separate query and key clustering, and integrating exponential tilting for pretrained model compatibility. The insight that query clustering improves monopole approximation quality by 4-5× while barely affecting retrieval selection is a novel contribution that deepens understanding of attention approximation.

---

> ### Author Rebuttal · Authors · 2026-03-30
>
> We thank Reviewer 5Eph for their detailed and technically engaged review.
>
> **Q1: Clustering overhead and scalability.** The cost of clustering is dominated by computing the similarity of all N tokens against C centroids, expressed as a matmul of size NCD. In terms of FLOPs, clustering requires 4 such matmuls, compared to 5 in the monopole forward pass and 12 in the monopole backward pass — clustering is under 20% of the monopole computation alone, before accounting for retrieval and block-diagonal exact attention. Crucially, clustering has the same O(NCD) scaling as the rest of the approximation and is strictly cheaper than the O(N^2 D) exact attention it replaces. From a FLOPs perspective, since the ratio between clustering and monopole cost is fixed, clustering cannot become a bottleneck as N or C increases.
>
> In terms of runtime at our 1B operating point (QK128R8, 64k context), the breakdown is: retrieval 37%, block-diagonal exact attention 30%, XLA fusions/sorts/transposes 18%, monopole 8.8%, clustering 8.4%. Currently, clustering is marginally cheaper than monopole, and both are small minorities of overall cost. As expected from the previous analysis, clustering remains at 7-8% of total runtime as cluster count doubles from 64 to 128 to 256 at constant retrieval fraction. Clustering has not been a major optimization target given this profile, and has significant optimization headroom remaining.
>
> **Q2: Retrieval robustness.** We first clarify: MuSe is stateless — clustering and retrieval selection are recomputed from the actual Q/K values on every forward pass, with no memory between sequences. Additionally, MuSe is off at inference (models use exact attention). There is therefore no notion of attention patterns "unseen during training" — MuSe adapts to the Q/K geometry independently per-head on each forward pass during training, and is replaced by exact attention at inference.
>
> Regarding the question of whether fixed k1/k2 parameters can miss important information when attention is dispersed: the potential failure mode is real but extremely narrow. Retrieval is independent per token and per head, so different tokens or heads needing spatially distant information is not a problem. The failure case therefore requires a single token on a single head needing fine-grained access to specific keys from locations separated by whole documents, falling in different spatial blocks of the same semantic cluster. Moreover, simple lookup (where any copy of the relevant key suffices) is robust to spatial dispersion — only *aggregation* across non-redundant keys in different spatial blocks of the same cluster poses a problem. Were this significant, we would expect increasing k2 (spatial blocks per cluster) while reducing k1 (number of clusters) at constant retrieval budget to improve performance. This is why we separate k1 and k2 instead of using a single retrieval parameter, and why our implementation allows the ratio to be tuned. We found that semantic breadth dominated spatial breadth on our data at our operating point. On a domain where these specific spatially dispersed aggregations turned out to be important, shifting budget from k1 to k2 would be the natural adjustment. Results are below (first line is our operating point, corresponding to the bolded line of Table 2 in the main body):
>
> | Config | Clusters (R) | Spatial (SP) | Total pairs | RSE | Corr |
> |--------|-------------|-------------|-------------|------|------|
> | R8SP1 | 8 | 1 | 8 | 0.00622 | 0.9965 |
> | R4SP2 | 4 | 2 | 8 | 0.00703 | 0.9959 |
> | R2SP4 | 2 | 4 | 8 | 0.01797 | 0.9892 |
>
>
> **Head dimension.** Like most comparable works, we focus on a primary choice of head dimension. Our choice (64) follows recent frontier practice (OpenAI GPT-OSS 20B/120B) and is natural for our 96M-1B scaling range. We additionally validate head dim 128 via both pretraining (Table 7) and Llama 3.1-8B generalization experiments.
>
> **Downstream evaluation.** We have conducted additional long-context retrieval evaluations (RULER NIAH, custom code NIAH, LongBench code completion) at 64k. MuSe-trained models match or outperform exact-attention-trained models — see our response to Reviewer AkdV for details.
>
> **Dimension scores.** We note that your prose describes the paper as "technically sound" with "well-designed experiments and thorough ablation studies" (Soundness), demonstrating "strong originality" with a "novel contribution" (Originality), and addressing an "important and timely problem" with "meaningful practical gains" (Significance), while the corresponding dimension scores are 2/Fair. We wanted to flag this in case it does not reflect your intended evaluation. We would welcome any guidance on remaining concerns.

---

> > ### Author Rebuttal · Reviewer_5Eph · 2026-04-08
> >
> > I will keep the score and recommend acceptance.

---

### Official Review · Reviewer_AkdV · 2026-03-12

**Soundness:** 3
**Presentation:** 3
**Significance:** 3
**Originality:** 3
**Overall Recommendation:** 4
**Confidence:** 3

**Summary:**

The paper introduces Multipole Semantic Attention (MuSe), an efficient approximation of softmax attention designed to accelerate transformer pretraining on long sequences. The method addresses the quadratic computational cost of standard attention, which becomes a bottleneck when training models with long context windows such as 64k tokens.

**Compliance With Llm Reviewing Policy:**

Affirmed.

**Final Justification:**

The authors have addresses my concerns, I have revised my rating.

**Key Questions For Authors:**

1.	Long-Range Reasoning: How does MuSe perform on standard long-context retrieval benchmarks, such as the "Needle In A Haystack" test?
2.	Fine-tuning Necessity: The paper mentions that shifting from MuSe to Exact Attention requires a 0.1% fine-tuning phase. Does this imply a persistent bias in the representation space learned by MuSe?

**Limitations:**

yes

**Strengths And Weaknesses:**

Strengths
1. Well-motivated design. The proposed method combines semantic clustering with multipole-inspired approximations to reduce the cost of attention. The two-stage mechanism (cluster-level summarization followed by token-level refinement) provides a reasonable trade-off between computational efficiency and approximation accuracy.
2. Addresses an important practical problem. The quadratic complexity of attention remains a major bottleneck for long-context transformer training. Achieving a reported 36% training speedup at 64k context without architectural changes or retraining makes the approach practically appealing.
3. Reasonable empirical evaluation. The paper includes both microbenchmark results and end-to-end pretraining experiments up to 1B parameters. Experiments on realistic datasets such as code repositories and scientific documents support the claim that the method is applicable to practical workloads.
4. Interesting perspective on attention approximation. Although clustering-based approaches are not entirely new, the idea of independently clustering queries and keys and combining hierarchical summaries with retrieval introduces an interesting perspective compared to typical sparse or low-rank approximations.

Weaknesses
1. Limited pretraining scale. The pretraining experiment focuses on models up to 1B parameters and 64k context. It remains unclear whether the method scales effectively to larger models (e.g., 7B+) when pretraining from scratch, or longer contexts typical of modern large language models.
2. Narrow Long-Context Evaluation: The paper relies heavily on Language Modeling (LM) loss and perplexity to justify long-context capabilities. However, a lower loss does not necessarily translate to functional long-range reasoning. The absence of specialized benchmarks—such as Needle In A Haystack, LongBench, or RULER—makes it hard to assess if the semantic clustering preserves fine-grained retrieval and reasoning across 64k+ tokens.
3. Limited Downstream Task Coverage: The evaluated downstream tasks (ARC-Easy and SciQ) are relatively short-context. These do not effectively stress-test the core contribution of the paper, which is long-context modeling.

---

> ### Author Rebuttal · Authors · 2026-03-30
>
> We thank Reviewer AkdV for their thoughtful assessment and for highlighting long-context retrieval evaluation as a valuable perspective. We address each concern below.
>
> **Clarification on evaluation setting.** MuSe is a training-time approximation only. At inference, all models use exact (CUDNN) attention. All evaluations below are conducted with exact attention in the intended deployment setting. The question is therefore not how MuSe attention affects long-range reasoning at inference, but whether training with MuSe produces models that reason well over long contexts when exact attention is restored. The full capabilities of exact attention are available at inference — the model merely needs to have learned to use them.
>
> **Long-context retrieval evaluation (W2, W3, Q1).** We agree that explicitly evaluating the retrieval component enriches the existing evidence, and have conducted three sets of long-context evaluations on our code models. We emphasize code as a particularly demanding retrieval domain: function definitions must be retrieved via imports across long distances, variable bindings span large scopes, and cross-file dependencies are pervasive.
>
> *RULER at 64k (500 samples).* On single-needle retrieval (niah_single_1), MuSe outperforms CUDNN on both domains: 85% vs 61% (PDF) and 71% vs 63% (code). On more complex RULER NIAH variants, the two domains diverge sharply: the PDF model shows moderate scores with CUDNN ahead (e.g., 47% vs 27% on niah_single_2), while the code model scores near zero on the same tasks regardless of training method (e.g., 0% vs 1% on niah_single_2). This divergence — a task going from 47% to 0% between domains — indicates that complex RULER variants are probing domain-specific format compatibility rather than isolating retrieval capability, motivating the in-distribution evaluation below. Non-NIAH subtasks (VT, CWE, FWE, QA) produce equivalent or near-zero scores for all models.
>
> *Custom code NIAH at 64k (288 sequences, 9 depth buckets).* We therefore replicate RULER's single-needle, multi-key, and multi-query task structure using code-native cross-file function imports, which serve as a natural retrieval primitive in this domain. A file containing a function definition (key) is inserted into the repository; a separate file imports and calls that function (query). The function name contains a random uid (value), so correct completion requires retrieval rather than guessing. We evaluate K=2,4 multi-key (target among distractors with distinct filenames and function names) and N=2,4 multi-query (multiple needles at different depths, all probed) variants alongside the single-needle baseline. Results (uid-level exact match):
>
> | Task | CUDNN | MuSe |
> |------|-------|------|
> | Single-needle | 97.8% | 98.4% |
> | Multi-key K=2 | 92.7% | 98.5% |
> | Multi-key K=4 | 81.5% | 97.4% |
> | Multi-query N=2 | 88.5% | 98.7% |
> | Multi-query N=4 | 83.6% | 98.8% |
>
> MuSe maintains above 90% at all depths with no positional degradation pattern, while CUDNN performance varies substantially with needle depth on complex variants (e.g., 64–97% across positions on multi-key K=4). Notably, MuSe matches or exceeds CUDNN on all variants.
>
> *LongBench code (RepoBench-P and LCC).* We evaluate the code-completion subset of LongBench, which is in-distribution for our models (remaining LongBench tasks require instruction following). MuSe matches exact attention: RepoBench-P edit similarity 0.393 vs 0.382, LCC 0.314 vs 0.329.
>
> **Scale (W1).** From 185M to 1B, MuSe and exact attention follow indistinguishable scaling laws (loss ∝ params^{-0.17}, R² > 0.999 for both). This directly tests whether the approximation introduces scale-dependent degradation, and finds none — a stronger signal than a single large run would provide. Our Llama 3.1-8B evaluation confirms that the approximation remains effective at 8B scale. At 64k context, a 7B+ comparison at chinchilla-optimal tokens would require ~65,000 A100-hours per domain.
>
> **Adaptation / fine-tuning (Q2).** The adaptation is shallow: 0.1% of pretraining tokens (26M) removes it entirely, and the fine-tuned model beats the baseline (0.6994 vs 0.7026). We have since identified the specific component responsible — gradients flowing through the softmax aggregation weights in the monopole summarization stage — and found that treating these weights as non-differentiable eliminates the adaptation entirely, with the MuSe-trained model beating the baseline on code without any fine-tuning (0.6991 vs 0.7026).
>
> Importantly, the adaptation does not affect retrieval: the non-fine-tuned model achieves uid_EM within 1% of the fine-tuned model on all NIAH tasks (e.g., 98.0% vs 98.8% on multi-query N=4), confirming it is limited to soft distributional features — consistent with the gradient pathway being in the monopole component.
>
> We thank the reviewer for suggesting the retrieval evaluation perspective, which we believe complements and enriches the existing training quality evidence.

---

> > ### Author Rebuttal · Reviewer_AkdV · 2026-04-02
> >
> > The authors have addresses my concerns, I have revised my rating.

---

### Official Review · Reviewer_xkjZ · 2026-03-13

**Soundness:** 3
**Presentation:** 3
**Significance:** 3
**Originality:** 3
**Overall Recommendation:** 4
**Confidence:** 3

**Summary:**

The paper proposes Multipole Semantic Attention (MuSe), an efficient approximation of softmax attention for accelerating long-context transformer pre-training. MuSe clusters queries and keys separately and computes cluster summaries to approximate full attention interactions, and retrieves a small number of cluster-block pairs for exact attention computation to further preserve accuracy. Experiments show that MuSe achieves significant training throughput improvements while maintaining comparable training loss to exact attention baselines.

**Compliance With Llm Reviewing Policy:**

Affirmed.

**Key Questions For Authors:**

See weaknesses.

**Limitations:**

Yes.

**Strengths And Weaknesses:**

Strengths:
* The paper is well-motivated and well written, addressing the quadratic cost of attention in long-context pre-training.
* The clustering of both queries and keys is technically sound and results in accurate attention approximation and competitive training loss empirically.
* The combination of semantic clustering with a two-level retrieval mechanism is intuitive and effectively balances approximation and exact computation.
* The evaluation is comprehensive, including microbenchmarks and end-to-end pertaining experiments, with good ablations.

Weaknesses:
* The evaluation is mostly limited to training loss and approximation error. Although the authors did evaluate two tasks, it would be valuable to evaluate the method on more challenging long-context tasks such as long-horizon reasoning or long-context retrieval benchmarks to better demonstrate its effectiveness in practical long-context applications.
* The largest model evaluated is 1B parameters, which is relatively small compared to modern LLM pre-training scales, leaving open the question of whether the method scales equally well to larger models.

---

> ### Author Rebuttal · Authors · 2026-03-30
>
> We thank Reviewer xkjZ for their positive assessment, and in particular for recognizing the technical soundness of our clustering approach and the comprehensiveness of our evaluation and ablations.
>
> **Long-context evaluation.** We emphasize that MuSe is a training-time approximation; all models deploy with exact attention, so any capability gap must arise from the training signal, not the inference mechanism.
>
> Following your suggestion, we have conducted long-context retrieval and downstream evaluations at 64k on our code models — a domain where retrieval is central (imports, cross-file references, long-range variable dependencies).
>
> On RULER single-needle NIAH, MuSe outperforms CUDNN (85% vs 61% on PDF, 71% vs 63% on code). On more complex RULER variants, CUDNN leads on some PDF tasks (e.g., 50% vs 28% on multikey retrieval), but the same tasks score near zero on code models regardless of training method, suggesting these variants probe domain-specific formatting or instruction following rather than retrieval per se. To isolate retrieval in-distribution, we designed a code-native NIAH evaluation with single-needle, multi-key (K=4, target among distractors), and multi-query (N=4) variants. MuSe achieves 97–99% uid-level exact match across all variants, versus 82–98% for CUDNN, with CUDNN showing substantial positional variation on complex variants (e.g., 64-97% across depth buckets on multi-key K=4). On the code-related subset of LongBench (RepoBench-P and LCC), MuSe matches exact attention. Full details are provided in our response to Reviewer AkdV.
>
> **Scale.** From 185M to 1B, MuSe and exact attention follow indistinguishable scaling laws (loss ∝ params^{-0.17}, R² > 0.999 for both), directly testing for scale-dependent degradation and finding none. We note that 1B-parameter pretraining at 64k context represents a substantial compute investment (8x attention flops per token vs the 8k context of most comparable methods), comparable to the largest from-scratch pretraining in related work. Our Llama 3.1-8B evaluation additionally confirms that the approximation remains effective at 8B scale.

---

> > ### Author Rebuttal · Reviewer_xkjZ · 2026-04-02
> >
> > I will keep the score and recommend acceptance.

---

### Decision · Program_Chairs · 2026-04-30

**Decision:**

Accept (regular)

**Comment:**

The paper introduces Multipole Semantic Attention (MuSe), a novel approximation for softmax attention aimed at mitigating the quadratic scaling bottleneck during long-context transformer pretraining. It addresses a critical bottleneck in modern LLM training. All reviewers recognize that it is technically solid, clearly written, and provides a significant practical contribution to the field of efficient deep learning. While early reviews raise some concerns, the rebuttal phase successfully addressed all of them, leading to a consensus that the work is of high quality and ready for publication.